# Self-Discriminative Modeling for Anomalous Graph Detection

Jinyu Cai [* 1]   Yunhe Zhang [* 2]   Jicong Fan [3]

## Abstract

Identifying anomalous graphs is essential in real-world scenarios such as molecular and social network analysis, yet anomalous samples are generally scarce and unavailable. This paper proposes a Self-Discriminative Modeling (SDM) framework that trains a deep neural network only on normal graphs to detect anomalous graphs. The neural network simultaneously learns to construct pseudo-anomalous graphs from normal graphs and learns an anomaly detector to recognize these pseudo-anomalous graphs. As a result, these pseudo-anomalous graphs interpolate between normal graphs and real anomalous graphs, which leads to a reliable decision boundary of anomaly detection. In this framework, we develop three algorithms with different computational efficiencies and stabilities for anomalous graph detection. Extensive experiments on 12 different graph benchmarks demonstrated that the three variants of SDM consistently outperform the state-of-the-art GLAD baselines. The success of our methods stems from the integration of the discriminative classifier and the well-posed pseudo-anomalous graphs, which provided new insights for graph-level anomaly detection.

## 1. Introduction

Graphs are widely utilized to represent complex relationships or interactions between entities in a variety of real-world contexts, such as molecules, biology, and social network data analysis (Mislove et al., 2007; Li et al., 2021; Sun et al., 2023; 2024). Although there have been a lot of works of anomaly detection on image data (Cai & Fan, 2022; Fu

*Equal contribution [1]Institute of Data Science, National University of Singapore, Singapore [2]Department of Computer and Information Science, SKL-IOTSC, University of Macau, Macau, China [3]School of Data Science, The Chinese University of Hong Kong, Shenzhen, China. Correspondence to: Jicong Fan <fanjicong@cuhk.edu.cn>.

*Proceedings of the 42nd International Conference on Machine Learning*, Vancouver, Canada. PMLR 267, 2025. Copyright 2025 by the author(s).

et al., 2024), tabular data (Dai et al., 2025; Dai & Fan, 2025), and graph nodes or edges (Duan et al., 2020; Zheng et al., 2021), graph-level anomaly detection (GLAD) (Akoglu et al., 2015; Qiao et al., 2024a), which aims to identify entire graphs that substantially deviate from normal patterns, remains a greater challenge due to the difficulty in analyzing overall relationship between graphs.

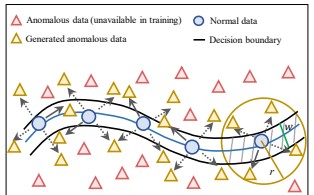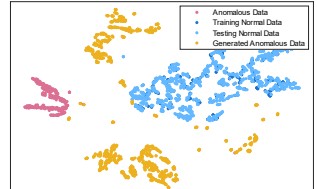

*Figure 1.* Motivation of our method. The left plot illustrates a toy example, where the shadowed region is approximated as a rectangle (or a hypercuboid when $d > 3$). The right plot shows a real example, the t-SNE (Van der Maaten & Hinton, 2008) visualization of the result of our method on the real dataset AIDS.

Early research on GLAD explored the graph kernel (GK) methods (Vishwanathan et al., 2010; Shervashidze et al., 2011) to identify graph anomalies by utilizing various similarity measurements, though these approaches are limited by their scalability. The emergence of graph neural networks (GNNs) (Kipf & Welling, 2017; Xu et al., 2019; Huang et al., 2023) has significantly advanced the development of GLAD through its powerful graph representation learning capability. Recent works (Zhao & Akoglu, 2021; Qiu et al., 2022; Zhang et al., 2024) have explored the integration of GNNs with deep one-class classification (DeepSVDD) (Ruff et al., 2018) for GLAD. A hypersphere is taught to model the distribution of normal graphs in the latent space, and anomalies are identified as graphs whose embeddings fall outside of the hypersphere. Beyond these methods, other works have leveraged various paradigms, such as knowledge distillation (Ma et al., 2022), information bottleneck (Liu et al., 2023a), and Rayleigh quotient (Dong et al., 2024), to enhance the detection of anomalies. Moreover, semi-supervised learning (Zhang et al., 2022a) has been investigated to address the imbalance issue in GLAD by incorporating a small fraction of labeled anomalies for decision boundary learning.

Despite the recent advances, several critical challenges still remain to be addressed. The DeepSVDD-based ap-

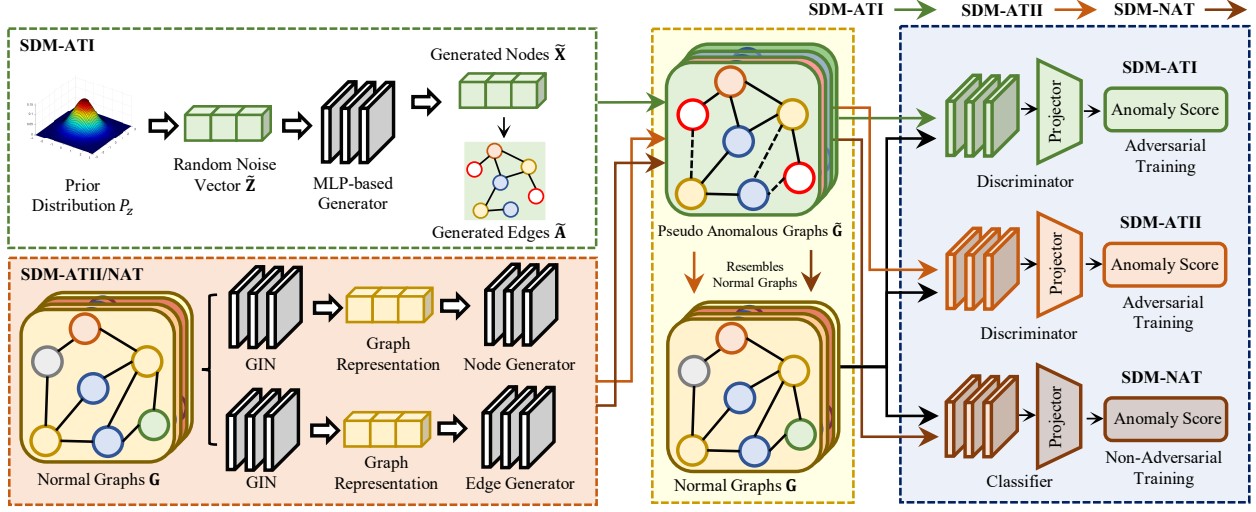

*Figure 2.* Illustration of the architecture of the proposed SDM variants. The **Green**, **Orange**, and **Brown** arrows denote the workflows of the SDM-ATI, SDM-ATII, and SDM-NAT, respectively.

proaches (Zhao & Akoglu, 2021; Qiu et al., 2022; Zhang et al., 2024) rely on the strong assumption that the distribution of graph embeddings follows a hypersphere, which may not be realistically achievable in real-world scenarios. Additionally, some approaches (Ma et al., 2022; Liu et al., 2023a; Dong et al., 2024) require explicitly defined anomaly scores, which can be challenging to apply in practice as the criteria for measuring anomalous graphs could vary significantly across datasets. Although semi-supervised approaches (Zhang et al., 2022a) are a promising solution, their effectiveness is constrained by the limited diversity of available graph anomalies in real-world scenarios.

In this paper, we propose Self-Discriminative Modeling (SDM), a novel framework for graph-level anomaly detection. The key idea (illustrated in Figure 1) of SDM is to distinguish normal graphs from the generated pseudo-anomalous graphs that interpolate between normal and (real) anomalous ones. We introduce two approaches to generate such pseudo-anomalous graphs: (1) training a generator using random noise from a latent distribution and (2) training a perturbator to create anomalies from normal graphs. Both approaches leverage adversarial training, incorporating a discriminator to differentiate between normal and pseudo-anomalous graphs. Moreover, we propose a non-adversarial variant of SDM to enhance model stability and accuracy. Based on each of the three approaches, the discriminator/classifier serves as the anomaly detector and adaptively learns the decision boundary between normality and abnormality. Figure 2 presents the network structure of SDM (three variants). Our contributions are as follows:

- We propose a novel and efficient GLAD framework that revolves around training a discriminator (classifier) to distin-

guish normal graphs from well-posed pseudo-anomalous graphs more effectively.

- We introduce two adversarial approaches to produce pseudo-anomalous graphs that closely resemble normal graphs but are more similar to anomalous ones, where a discriminator is learned jointly.

- We introduce a non-adversarial approach, which generates pseudo-anomalous graphs by adaptively perturbing normal graphs, and trains a classifier to distinguish them.

- We demonstrate the effectiveness of three variants of SDM against the state-of-the-art GLAD methods on 12 graph benchmark datasets.

## 2. Self-Discriminative Modeling for Anomalous Graph Detection

### 2.1. Problem Formulation and Motivation Description

**Problem Formulation.** Let $\mathbb{G} = \{G_1, \ldots, G_N\}$ be a graph dataset comprising $N$ graphs, where a single graph $G_i = \{V_i, E_i\}$ contains a node set $V_i$ and an edge set $E_i$. The adjacency matrix of $G_i$ is represented by $\mathbf{A}_i \in \{0,1\}^{n_i \times n_i}$, where $n_i = |V_i|$. The feature matrix of nodes of $G_i$ is denoted as $\mathbf{X}_i \in \mathbb{R}^{n_i \times d}$. Suppose the graphs in $\mathbb{G}$ are normal graphs, we want to train a model from $\mathbb{G}$ to determine whether a new graph $G_{\text{new}}$ is normal or abnormal. This problem is called *anomalous graph detection* (AGD)[1]. A key assumption of the AGD problem is that $G_1, \ldots, G_N$

---

[1]Note that this is an unsupervised learning problem, of which the training data do not contain any anomalous graphs. There are also supervised and semi-supervised settings (Ruff et al., 2020; Zhang et al., 2022a).

are drawn from an unknown distribution $\mathscr{D}$ (deemed as a normal distribution), while any graphs drawn from any other distributions, denoted as $\tilde{\mathscr{D}}$, are anomalous. Very importantly, there is no overlap between $\mathscr{D}$ and all possible $\tilde{\mathscr{D}}$. The AGD problem can be regarded as a binary classification problem, *i.e.*, justifying $G \sim \mathscr{D}$ or $G \sim \tilde{\mathscr{D}}$. The objective is to train a classifier $f_\theta$, using only the given graph set $\mathbb{G}$, to distinguish between $G$ drawn from $\mathscr{D}$ and $\tilde{G}$ drawn from $\tilde{\mathscr{D}}$. However, the difficulty is that $\tilde{\mathscr{D}}$ is completely unknown. Therefore, we need to estimate $\tilde{\mathscr{D}}$ from $\mathbb{G}$, or at least generate some samples drawn from a subset of $\tilde{\mathscr{D}}$ using $\mathbb{G}$. To achieve this, we may solve the following problem

$$\underset{\theta, \tilde{\mathscr{D}}_s}{\text{minimize}} \quad \underset{G \sim \mathscr{D}}{\mathbb{E}} \ell(y, f_\theta(G)) + \underset{\tilde{G} \sim \tilde{\mathscr{D}}_s \subseteq \tilde{\mathscr{D}}}{\mathbb{E}} \ell(\tilde{y}, f_\theta(\tilde{G})),$$
$$\text{subject to } \text{dist}(\mathscr{D}, \tilde{\mathscr{D}}_s) \leq \tau, \tag{1}$$

where $y \equiv 1$ and $\tilde{y} \equiv 0$ denote the labels of normal and anomalous graphs, respectively, $f_\theta(\cdot)$ denotes a classifier (*e.g.* a neural network) parameterized with $\theta$, and $\ell(\cdot)$ denotes the loss function. The constraint in Eq. (1) means that $\mathscr{D}$ and $\tilde{\mathscr{D}}_s$ should be close enough with respect to a distance metric $\text{dist}(\cdot, \cdot)$, where $\tau > 0$ is a small constant. However, in Eq. (1), $\tilde{\mathscr{D}}$ remains unknown, and the requirement for $\tilde{\mathscr{D}}_s$ to not overlap with $\mathscr{D}$ can be overly restrictive. Even if $\tilde{\mathscr{D}}_s$ overlap with $\mathscr{D}$, the learned $f_\theta$ could still be effective, provided that the decision boundary encloses $\mathscr{D}$ compactly (to be shown in Figure 1).

Instead of solving Eq. (1), we propose to train a model $g_\phi$ that is able to generate pseudo-anomalous graphs from normal graphs and simultaneously train a classifier $f_\theta$ that can provide a reliable decision boundary between normal graphs and anomalous graphs. To ensure an effective detector $f_\theta$, we should guarantee the following properties:

• The generated pseudo-anomalous graphs by $g_\phi$ interpolate between normal ones and (real) anomalous ones.

• $f_\theta$ distinguishes between the normal training graphs and most of the generated pseudo-anomalous graphs.

We therefore solve

$$\underset{\theta, \phi}{\min} \quad \underset{G \sim \mathscr{D}}{\mathbb{E}} \ell(y, f_\theta(G)) + \underset{\tilde{G} \sim g_\phi(G), G \sim \mathscr{D}}{\mathbb{E}} \ell(\tilde{y}, f_\theta(\tilde{G})), \tag{2}$$

where $g_\phi(\cdot)$ converts a normal graph to a distribution of pseudo-anomalous graphs.

**Motivation Description.** Here we visualize our motivation and show that the two aforementioned properties can be guaranteed. As shown in Figure 1, the first plot summarizes the motivation of Eq. (2). Specifically, the blue points represent normal training data, roughly lying on a (blue) curve. $g_\phi$ perturbs each normal graph randomly to generate one or more pseudo-anomalous graphs. We see that most pseudo-anomalous graphs are far from the blue

curve, which can be theoretically proved as follows. Let's consider a more general case in $d$-dimension space. The shadowed region (between the two black curves in 2D) in the radius-$r$ hypersphere (the yellow circle in 2D) can be approximated as a hypercuboid. Therefore, the volume of the shadowed region is approximated by

$$V_S = \prod_{i=1}^{d} w_i, \tag{3}$$

where $w$ denotes the width of hypercuboid, $w_1 = \cdots = w_\alpha = 2r$ and $1 \leq \alpha < d$. Then, the ratio of expected numbers of pseudo-anomalous graphs in the shadowed region and the unshadowed region in the hypersphere is

$$\eta = \frac{(2r)^\alpha \prod_{i=\alpha+1}^{d} w_i}{\frac{\pi^{d/2} r^d}{\Gamma(1+d/2)} - (2r)^\alpha \prod_{i=\alpha+1}^{d} w_i}, \tag{4}$$

where we have, WLOG, assumed that the points are distributed uniformly because $\tilde{G} \sim g_\phi(G)$ is a stochastic operation. To simplify the analysis, we show the following two examples of Eq. (4):

$$\eta = \begin{cases} \frac{2w}{\pi r - 2w}, & \text{if } d = 2 \text{ and } \alpha = 1, \\ \frac{4wr^2}{4\pi r^3/3 - 4wr^2}, & \text{if } d = 3 \text{ and } \alpha = 2. \end{cases} \tag{5}$$

We see that $\eta$ decreases when $w$ decreases or $r$ increases, where $r$ is related to the variation of pseudo-anomalous graphs. We can conclude that most pseudo-anomalous graphs are outside the shadowed region when there are some small $w_i$, namely, the latent dimension of the normal data is much lower than the ambient dimension. It also means they interpolate between normal graphs and real anomalous graphs. In this case, a classifier that can distinguish between the normal training data and most pseudo-anomalous graphs is sufficient to detect anomalous graphs. Besides, in Eq. (2), $f_\theta$ should not be too complex. Otherwise, it will overfit on $\mathbb{G}$ and the pseudo-anomalous graphs, leading to poor generalization to unseen graphs.

The second plot in Figure 1 is a visualization of our method on a real dataset and highlights the successful learning of a useful decision boundary: the generated anomalous graphs surround the normal ones, alongside the (real) anomalous ones. More visualization results of real examples refer to Figures 7 and 8 (Appendix E). We call Eq. (2) **Self-Discriminative Modeling** (SDM) based GLAD. In the following three sections, we will show how to solve Eq. (2) approximately and develop three variants of SDM.

### 2.2. Self-Discriminative Modeling: SDM-ATI

We first present a GAN-based approach to generate pseudo-anomalous graphs. The model consists of a graph generator $\mathcal{G}_\phi$ and a graph discriminator $\mathcal{D}_\omega$, which are alternatively trained in an adversarial manner. The generator tries

to produce fake (pseudo-anomalous) graphs (containing nodes and edges generation) that can fool the discriminator, while the discriminator tries to differentiate between anomalous and normal graphs. Specifically, the generator $\mathcal{G}_\phi$ generates nodes and edges to form a fake graph set $\tilde{\mathbb{G}} = \{\tilde{G}_1, \ldots, \tilde{G}_N\}$. We first sample random variable $\tilde{\mathbf{Z}}$ from a latent distribution $\mathbb{P}_{\tilde{\mathbf{Z}}} := \mathcal{N}(\mathbf{0}, \mathbf{1})$ and construct the adjacency matrix as follows

$$\tilde{\mathbf{A}} = \mathcal{T}(\tilde{\mathbf{X}}\tilde{\mathbf{X}}^\top), \quad \tilde{\mathbf{X}} = \mathcal{G}_\phi(\tilde{\mathbf{Z}}), \quad \tilde{\mathbf{Z}} \sim \mathbb{P}_{\tilde{\mathbf{Z}}}, \qquad (6)$$

where $\mathcal{G}_\phi$ is a Multi-Layer Perceptron (MLP)-based generator that maps the random latent variable $\tilde{\mathbf{Z}} \in \mathbb{R}^{N \times d}$ to the anomalous node attributes, and $\mathcal{T} : \mathbb{R} \to [0, 1]$ denotes an element-wise transformation function, *e.g.* Sigmoid$(\cdot)$. In this way, we generate an anomalous graph set $\tilde{\mathbb{G}}$ with the generator $\mathcal{G}_\phi$. We then introduce a discriminator $\mathcal{D}_\omega$, which takes the anomalous graphs $\tilde{\mathbb{G}}$ and normal graphs $\mathbb{G}$ as input, and aims to distinguish between them effectively. To fully exploit the structural information of graphs, $\mathcal{D}_\omega$ is expected to be a GNN-based discriminator. Specifically, we leverage GIN (Xu et al., 2019) as the backbone network of the discriminator $\mathcal{D}_\omega$ to learn graph-level representations. Assume we have an input graph $G_i$, the latent features $\mathbf{h}^{(k)}(v)$ of node $v$ in the $k$-th layer of GIN can be obtained by aggregating the learned features from its neighboring nodes in the $(k-1)$-th layer, which can be formulated as

$$\begin{aligned} \mathbf{h}^{(k)}(v) = &\delta(\text{COMBINE}(\mathbf{h}^{(k-1)}(v), \\ &\text{AGGREGATE}(\{\mathbf{h}^{(k-1)}(u), u \in \mathcal{C}(v)\}))), \end{aligned} \qquad (7)$$

where $\mathcal{C}(v)$ denotes the neighbor set of node $v$, and $\delta(\cdot)$ is a non-linear activation function such as ReLU. $\text{AGGREGATE}(\cdot)$ function combines the features of neighboring nodes in $\mathcal{C}(v)$, and $\text{COMBINE}(\cdot)$ function combines the features from the previous layer and the aggregated neighborhood information to obtain the current layer's features. Note that the attribute $\mathbf{x}_v$ of node $v$ serves as the initial features, *i.e.*, $\mathbf{h}^{(0)}(v) = \mathbf{x}_v$. Then the graph-level representation of graph $G_i$ can be derived as follows

$$\mathbf{h}_{G_i} = \mathcal{R}(\{\text{CONCAT}(\mathbf{h}^{(k)}(v), k \in \{1, \ldots, K\})\}, v \in G_i), \quad (8)$$

where $\text{CONCAT}(\cdot)$ function concatenates the representations learned in each GIN layer, and $\mathcal{R}(\cdot)$ denotes the sum-readout function. Consequently, we can learn the graph-level representations $\mathbf{H}_\mathbb{G}$ and $\mathbf{H}_{\tilde{\mathbb{G}}}$ for normal and pseudo-anomalous graphs by aggregating the node features, and train the discriminator to distinguish them as much as possible. The generator $\mathcal{G}_\phi$ and discriminator $\mathcal{D}_\omega$ are alternatively optimized with a min-max game as follows

$$\begin{aligned} \min_\phi \max_\omega \; &\underset{\mathbf{X}_i, \mathbf{A}_i \sim \mathbb{P}_\mathbb{G}}{\mathbb{E}} [\mathcal{D}_\omega(\mathbf{X}_i, \mathbf{A}_i)] - \\ &\underset{\tilde{\mathbf{Z}}_i \sim \mathbb{P}_{\tilde{\mathbf{Z}}}}{\mathbb{E}} [\mathcal{D}_\omega(\mathcal{G}_\phi(\tilde{\mathbf{Z}}_i), \mathcal{T}(\mathcal{G}_\phi(\tilde{\mathbf{Z}}_i)\mathcal{G}_\phi(\tilde{\mathbf{Z}}_i)^\top))], \end{aligned} \qquad (9)$$

where $\mathbb{P}_\mathbb{G}$ denotes the normal graph data distribution, and $\tilde{\mathbf{Z}}_i \in \mathbb{R}^{n \times d'}$ is sampled from the prior distribution $\mathbb{P}_{\tilde{\mathbf{Z}}} \sim \mathcal{N}(\mathbf{0}, \mathbf{1})$. Particularly, the discriminator can naturally serve as an anomaly detector after training. Comparing to Eq. (1), we see that the constraint $\text{dist}(\mathscr{D}, \tilde{\mathscr{D}}_s) \leq \epsilon$ is guaranteed if $\mathcal{G}_\phi$ given by Eq. (9) is strong enough. It is difficult to guarantee for Eq. (1) that $\tilde{\mathscr{D}}_s$ does not overlap with $\mathscr{D}$, which however is not compulsory because it is still possible to learn a discriminator from overlapping $\mathscr{D}, \tilde{\mathscr{D}}_s$ to distinguish between $\mathscr{D}$ and $\tilde{\mathscr{D}}$. We call this method SDM-ATI, where AT represents adversarial training. Although SDM-ATI is promising for anomalous graph detection, it may suffer from the following problems:

- An MLP-based generator may not effectively capture the structural information of graphs, which could impede the generation of high-quality anomalous graphs.

- The interpretability of the GAN-based method is limited because generating anomalous graphs from random noise does not necessarily guarantee the quality of the generated anomalous graphs.

- The optimization of the GAN-based method involves a min-max game, which can lead to instability during training. Besides, the competition between the generator and discriminator may result in mode collapse, leading to the generation of poor-quality anomalous graphs.

## 2.3. Self-Discriminative Modeling: SDM-ATII

To address the first two problems of SDM-ATI, we propose a variant of our SDM-ATI, which can leverage the structural information, and further provide more explicit guidance for the generator $\mathcal{G}_\phi$, ensuring the generation of high-quality anomalous graphs that closely resemble normal ones but can still be distinguished by the discriminator. Specifically, we introduce the GIN-based VGAE (Kipf & Welling, 2016) framework to build the generator $\mathcal{G}_\phi$, which consists of a node generator and an edge generator to learn anomalous graphs. The node generator aims to generate anomalous attributes $\tilde{\mathbf{X}}$, while the edge generator, which does not include a decoder, aims to generate the adjacency matrix $\tilde{\mathbf{A}}$. Instead of sampling the input of $\mathcal{G}_\phi$ from the latent distribution $\mathbb{P}_{\tilde{\mathbf{Z}}}$, we take the normal graph set $\mathbb{G}$ as the input of $\mathcal{G}_\phi$, to generate anomalous graphs $\tilde{\mathbb{G}}$ that are close to $\mathbb{G}$ but are expressive pseudo-anomalous graphs.

Here, we only describe the node generator, as it differs from the edge generator just in the existence of a decoder. We first learns the graph-level representation $\mathbf{H}_\mathbb{G}$ for the input graphs $\mathbb{G} = \{G_1, \ldots, G_N\}$ by Eqs. (7) and (8), where $G_i = \{\mathbf{X}_i, \mathbf{A}_i\}$. Next, we map the graph-level representation into a latent Gaussian distribution $\mathcal{N}(\boldsymbol{\mu}, \boldsymbol{\sigma}^2)$ as in VGAE, where the means $\boldsymbol{\mu}$ and deviations $\boldsymbol{\sigma}$ are defined as

$$\boldsymbol{\mu} = \text{GIN}_{\boldsymbol{\mu}}(\mathbf{H}_\mathbb{G}, \mathbf{A}), \; \boldsymbol{\sigma} = \exp(\text{GIN}_{\boldsymbol{\sigma}}(\mathbf{H}_\mathbb{G}, \mathbf{A})), \quad (10)$$

where $\boldsymbol{\mu}$ and $\boldsymbol{\sigma}$ can explicitly define an inference model that we can sample the latent graph representations $\mathbf{Z}_{\mathbb{G}}$ from it as follows

$$q(\mathbf{Z}_{\mathbb{G}}|\mathbf{H}_{\mathbb{G}}, \mathbf{A}) = \prod_{i=1}^{N} q(\mathbf{Z}_{G_i}|\mathbf{H}_{\mathbb{G}}, \mathbf{A}), \qquad (11)$$

where $q(\mathbf{Z}_{G_i}|\mathbf{H}_{\mathbb{G}}, \mathbf{A}) = \mathcal{N}(\mathbf{Z}_{G_i}|\boldsymbol{\mu}_i, \mathrm{diag}(\boldsymbol{\sigma}_i))$. Since the sample operation could not provide gradient information, we leverage the reparametrization trick (Kingma & Welling, 2014) to sample the latent graph representation, *i.e.*,

$$\mathbf{Z}_{\mathbb{G}} = \boldsymbol{\mu} + \epsilon\boldsymbol{\sigma}, \ \epsilon \sim \mathcal{N}(\mathbf{0}, \mathbf{1}), \qquad (12)$$

where $\epsilon$ denotes the random Gaussian noise subject to the standard normal distribution. Consequently, we can generate a negative graph set, including edges and nodes by

$$\tilde{\mathbf{A}} = \mathcal{T}(\mathbf{Z}_{\mathbb{G}}\mathbf{Z}_{\mathbb{G}}^{\top}), \ \ \tilde{\mathbf{X}} = \mathrm{MLP}(\mathbf{Z}_{\mathbb{G}}), \qquad (13)$$

where $\mathrm{MLP}(\cdot)$ denotes an MLP-based decoder, which aims to generate anomalous attributes $\tilde{\mathbf{X}}$ from the latent graph representations, and the anomalous adjacent matrix $\tilde{\mathbf{A}}$ is generated from the latent graph representation learned by the edge generator following Eq. (13).

We expect to generate high-quality pseudo-anomalous graphs that closely resemble normal ones but can still be distinguished by the classifier. This requires a high level of similarity between the generated anomalous graphs and the normal graphs, which can be regarded as minimizing the discrepancy between the generated graphs and the normal ones. Therefore, we propose to minimize the following discrepancy loss:

$$
\begin{aligned}
L_{\mathrm{dis}} = \frac{1}{N} \sum_{i=1}^{N} \Big( &\left\| \mathbf{X}_i - \tilde{\mathbf{X}}_i \right\|_F^2 - \big( \mathbf{A}_i \log(\tilde{\mathbf{A}}_i) \\
&+ (1 - \mathbf{A}_i) \log(1 - \tilde{\mathbf{A}}_i) \big) \Big),
\end{aligned} \qquad (14)
$$

where $\tilde{\mathbf{X}}_i$ and $\tilde{\mathbf{A}}_i$ are the node attribute and adjacency matrix generated by the node generator and edge generator of $\mathcal{G}_{\phi}$ respectively, *i.e.*, $\tilde{\mathbf{X}}_i, \tilde{\mathbf{A}}_i = \mathcal{G}_{\phi}(\mathbf{X}_i, \mathbf{A}_i)$. The first term denotes the attribute reconstruction loss, and the second denotes the binary cross-entropy loss. Additionally, the distribution of learned latent representation $\mathbf{Z}_{\mathbb{G}}$ is expected to follow a pre-defined prior distribution, which allows the generated latent representations $\mathbf{Z}_{\mathbb{G}}$ to be uniformly distributed in the latent space, ensuring the diversity of generated graphs. We can achieve this by penalizing the KL-divergence between $q(\mathbf{Z}_{\mathbb{G}}|\mathbf{H}_{\mathbb{G}}, \mathbf{A})$ and a prior distribution $P(\mathbf{Z})$, *i.e.*, $KL[q(\mathbf{Z}_{\mathbb{G}}|\mathbf{H}_{\mathbb{G}}, \mathbf{A})||P(\mathbf{Z})]$, where $P(\mathbf{Z}) = \prod_i p(\mathbf{Z}_i) = \prod_i \mathcal{N}(\mathbf{Z}_i|\mathbf{0}, \mathbf{I})$ typically follows a Gaussian prior distribution. The overall objective function of the perturbation learning-based approach is

$$
\begin{aligned}
\min_{\phi} \max_{\omega} \ &\mathbb{E}_{\mathbf{X}_i, \mathbf{A}_i \sim \mathbb{P}_{\mathbb{G}}} [\mathcal{D}_{\omega}(\mathbf{X}_i, \mathbf{A}_i) - \mathcal{D}_{\omega}(\mathcal{G}_{\phi}(\mathbf{X}_i, \mathbf{A}_i))] \\
&+ \lambda L_{\mathrm{dis}} + \gamma KL[q(\mathbf{Z}_{\mathbb{G}}|\mathbf{H}_{\mathbb{G}}, \mathbf{A})||P(\mathbf{Z})],
\end{aligned} \qquad (15)
$$

where the discrepancy loss and KL-divergence terms are specific to the generator. This variant is based on adversarial training and perturbation learning, where the pseudo-anomalous graphs are generated via perturbing the latent variable of normal graphs. For convenience, we call this method SDM-ATII. Compared to the GAN-based method, SDM-ATI and SDM-ATII offer better interpretability by explicitly guiding the generator to generate pseudo-anomalous graphs that closely resemble the normal ones. Additionally, SDM-ATII offers better control over the diversity of the generated graphs by penalizing the KL-divergence between the distribution of learned latent graph representation and a prior Gaussian distribution. Compared to Eq. (2), we explicitly defined the discrepancy loss, *i.e.*, Eq. (14), to guarantee that the generated anomalous graphs surrounded the normal ones and learn the decision boundary from the adversarial training of the generator and discriminator. This variant offers improved interpretability compared to SDM-ATI, which relies solely on adversarial training between the generator and discriminator to guarantee the constraint. However, SDM-ATII still suffers from the instability of the min-max optimization.

## 2.4. Self-Discriminative Modeling: SDM-NAT

To address the instability of the min-max optimization in the adversarial approaches SDM-ATI and SDM-ATII, we further propose a non-adversarial variant called SDM-NAT for the perturbation learning-based variant SDM-ATII, which avoids the instability problem of GANs and simplifies the training process. Specifically, rather than training a generator and a discriminator to compete against each other, we proposed to directly train a classifier $f_{\theta}$ to distinguish the anomalous graphs produced by generator $\mathcal{G}_{\phi}$ from normal ones. We can accomplish this by utilizing the node generator and edge generator to produce a set of anomalous graphs $\tilde{\mathbb{G}}$ from normal graphs $\mathbb{G}$, and subsequently train a classifier to distinguish them. The overall objective is

$$
\begin{aligned}
\min_{\theta, \phi} \ \frac{1}{N} \sum_{i=1}^{N} \big( &\ell(y_i, f_{\theta}(\mathbf{X}_i, \mathbf{A}_i)) + \ell(\tilde{y}_i, f_{\theta}(\mathcal{G}_{\phi}(\mathbf{X}_i, \mathbf{A}_i))) \big) \\
&+ \lambda L_{\mathrm{dis}} + \gamma KL[q(\mathbf{Z}_{\mathbb{G}}|\mathbf{H}_{\mathbb{G}}, \mathbf{A})||P(\mathbf{Z})],
\end{aligned} \qquad (16)
$$

where $\ell(\cdot)$ denotes the binary cross-entropy loss of the classifier, the definitions of $L_{\mathrm{dis}}$ and the KL-divergence are exactly the same as Eq. (15). The classifier is based on GIN, which receives attribute and adjacency matrices as inputs, allowing for consideration of the structural information of the graphs. More importantly, our method is unsupervised, requiring no supervised information. We simply set $\tilde{y}_i = \cdots = \tilde{y}_N = 0$ for the generated anomalous graphs, and $y_1 = \cdots = y_N = 1$ for normal graphs. Compared to Eq. (2), we directly learn the decision boundary by simultaneously training a classifier with a generator that produces high-quality pseudo-anomalous graphs for the

classifier. This makes our method particularly appealing for real-world applications where obtaining labeled data is challenging and costly.

We further highlight the conceptual advances of our SDM framework by extensively discussing existing GLAD methods and GAN-based strategies in **Appendix** A. To facilitate the understanding of the proposed SDM methods, we also provide the detailed training procedures of SDM-ATI, SDM-ATII, and SDM-NAT in **Appendix** I.

# 3. Experiment

## 3.1. Experiment Configuration

**Datasets.** We evaluate the anomaly detection performance across various graph benchmark datasets, including (1) **Small and Moderate-scale Dataset:** Four molecule datasets (MUTAG, AIDS, COX2, and ER_MD), three biological datasets (PROTEINS, DD, and ENZYMES), and one social network dataset (IMDB-BINARY). These three types of data are typical graph-structured data in real-world scenarios. (2) **Imbalanced Large-scale Dataset:** Four imbalanced large-scale molecule graph datasets (SW-620, MOLT-4, PC-3, and MCF-7) to evaluate the GLAD performance under imbalanced scenarios. Note that all the datasets used in our experiment are sourced from TUDataset (Morris et al., 2020), a publicly available graph database[2]. For more details for each dataset, please refer to **Appendix** B.

**Network Architecture.** For the network architecture of the proposed SDM variants, we utilize a 3-layer GIN as the backbone network for the generator and discriminator (classifier) in SDM-ATII and SDM-NAT, except for the generator of SDM-ATI, which is an MLP-based neural network. The aggregated dimension and the dimensions of the two latent layers in GIN are set to 16 and 10, respectively.

**Baselines.** We extensively compare the three SDM variants with state-of-the-art GLAD baselines:

1. **Graph Kernel:** Short-Path kernel (SP) (Borgwardt & Kriegel, 2005), Weisfeiler-Lehman kernel (WL) (Shervashidze et al., 2011), Neighborhood Hash kernel (NH) (Hido & Kashima, 2009), and Random Walk kernel (RW) (Vishwanathan et al., 2010).

2. **GNN-based GLAD:** VGAE-AD (Kipf & Welling, 2016), GCN (Kipf & Welling, 2017), GIN (Xu et al., 2019), SOPOOL (Wang & Ji, 2020), RWGNN (Nikolentzos & Vazirgiannis, 2020), OCGIN (Zhao & Akoglu, 2021), GLocalKD (Ma et al., 2021), OCGTL (Qiu et al., 2022), iGAD (Zhang et al., 2022a), SIGNET (Liu et al., 2023a), MUSE (Kim et al., 2024), DO2HSC (Zhang et al., 2024).

[2]https://chrsmrrs.github.io/datasets/docs/datasets/

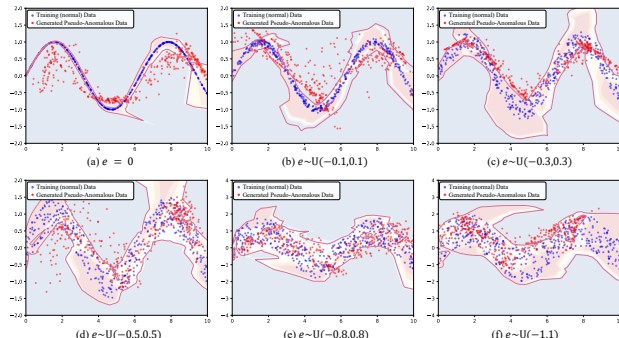

*Figure 3.* The decision boundaries learned from 2-D synthetic data $x = \sin(z) + e$ with different $e$.

**Evaluation Metrics.** We utilize AUC and F1-Score to evaluate the anomaly detection performance and report the mean value and standard deviation of 10 trials of each method on each dataset.

**Other Details.** We provide other settings in **Appendix** C, including the details regarding data split, trade-off parameters, baseline settings, training details, and implementation.

## 3.2. Simulation Analysis for Decision Boundary

We ran a simulation to show the key idea and effectiveness of our methods in addition to Figure 2. For convenience, we only consider SDM-NAT, and we will not use graphs because it is difficult to conduct a reasonable simulation for a number of graphs. Therefore, the corresponding backbones of SDM-NAT are changed to VAE (generator) and a common MLP-based classifier. Specifically, we generate a number of synthetic 2-D samples (normal training data) using $x = \sin(z) + e$, where $e$ is a noise drawn from a uniform distribution $(-a, a)$. A larger $a$ leads to a wider normal region. The simulation results are shown in Figure 3, where the pink curve denotes the learned decision boundaries, and the background color in the figure (changing from blue to red) implies that the score given by the classifier is increasing. The observations are as follows:

• The generated pseudo-anomalous data is usually distributed close to the training (normal) data and shows a similar manifold trend as the normal one.

• In most cases, the classifier can distinguish those pseudo-anomalous data far from the normal area.

• When the interval of normal data turns from narrow to wide, some generated pseudo-anomalous data may be located close to training data, but the classifier would neglect most of them and draw a superior decision boundary surrounding all training data.

In conclusion, the proposed model can effectively handle

*Table 1.* Average AUCs and F1-Scores with standard deviation (10 trials) of different GLAD methods. We assess models by regarding the individual classes as the normal class, respectively. The best results are highlighted in **bold** and "OM" means out of memory.

| Method | Metric | PROTEINS | | DD | | IMDB-BINARY | | ER_MD | |
|---|---|---|---|---|---|---|---|---|---|
| | | 0 | 1 | 0 | 1 | 0 | 1 | 0 | 1 |
| SP (Borgwardt & Kriegel, 2005) | AUC | 66.83±0.00 | 52.02±0.00 | 82.73±0.00 | 31.57±0.00 | 45.92±0.00 | 47.16±0.00 | 40.92±0.00 | 66.20±0.00 |
| | F1-Score | 63.64±0.00 | 44.50±0.00 | 76.09±0.00 | 38.14±0.00 | 59.00±0.00 | 61.00±0.00 | 37.74±0.00 | 63.89±0.00 |
| WL (Shervashidze et al., 2011) | AUC | 73.19±0.00 | 50.19±0.00 | 81.57±0.00 | 19.22±0.00 | 51.57±0.00 | 46.07±0.00 | 45.71±0.00 | 72.11±0.00 |
| | F1-Score | 76.82±0.00 | 33.80±0.00 | 74.64±0.00 | 26.80±0.00 | 61.00±0.00 | 58.00±0.00 | 45.28±0.00 | 69.44±0.00 |
| NH (Hido & Kashima, 2009) | AUC | 68.28±0.00 | 55.51±0.00 | 81.61±0.32 | 36.84±0.00 | 53.21±0.00 | 46.52±0.00 | 51.55±2.00 | 72.38±0.87 |
| | F1-Score | 68.79±0.23 | 41.93±0.21 | 73.91±0.65 | 40.82±1.54 | 57.71±1.46 | 64.40±0.80 | 50.19±0.92 | 63.89±0.00 |
| RW (Vishwanathan et al., 2010) | AUC | OM | OM | OM | OM | 49.51±0.00 | 53.11±0.00 | 78.94±0.00 | 70.33±0.00 |
| | F1-Score | OM | OM | OM | OM | 48.88±0.00 | 45.87±0.00 | 65.96±0.00 | 62.60±0.00 |
| VGAE-AD (Kipf & Welling, 2016) | AUC | 27.95±3.03 | 86.09±2.07 | 79.01±14.64 | 64.95±5.34 | 65.36±0.78 | 67.22±3.49 | 87.99±3.86 | 59.48±6.94 |
| | F1-Score | 35.15±2.17 | 79.33±2.06 | 70.42±7.55 | 76.67±0.39 | 62.40±3.76 | 64.79±4.35 | 81.51±4.20 | 58.89±5.39 |
| OCGIN (Zhao & Akoglu, 2021) | AUC | 55.01±9.65 | 47.77±7.64 | 66.59±4.44 | 60.03±5.34 | 34.82±1.67 | 65.04±8.79 | 47.63±3.59 | 67.51±1.32 |
| | F1-Score | 64.18±0.15 | 43.42±0.00 | 56.12±0.00 | 62.24±0.00 | 40.50±1.50 | 63.50±8.50 | 50.94±1.89 | 59.46±0.00 |
| GLocalKD (Ma et al., 2022) | AUC | 72.12±0.08 | 74.80±0.12 | 80.59±0.00 | 79.96±0.01 | 53.83±1.24 | 53.34±0.06 | 78.94±0.00 | 71.54±0.00 |
| | F1-Score | 70.76±0.37 | 71.78±0.54 | 73.48±0.57 | 71.13±0.00 | 54.91±0.33 | 56.06±0.11 | 70.21±0.00 | 62.86±0.00 |
| OCGTL (Qiu et al., 2022) | AUC | 63.20±5.40 | 58.10±6.10 | 77.52±0.43 | 82.45±0.19 | 65.10±1.80 | 64.12±1.27 | 72.67±0.20 | 29.63±0.18 |
| | F1-Score | N/A | N/A | 71.65±0.73 | 47.96±2.14 | 36.48±1.56 | 61.58±1.21 | 67.17±0.92 | 53.65±1.77 |
| SIGNET (Liu et al., 2023a) | AUC | 69.51±0.76 | 73.58±0.85 | 59.53±3.45 | 59.92±0.70 | 67.9±0.96 | 64.31±0.44 | 75.18±8.12 | 60.43±7.33 |
| | F1-Score | 64.39±0.62 | 73.33±0.91 | 56.76±3.47 | 59.11±0.49 | 64.67±0.47 | 62.33±0.47 | 60.43±7.33 | 56.11±6.43 |
| MUSE (Kim et al., 2024) | AUC | 73.17±1.20 | 41.86±1.31 | 61.06±3.03 | 38.01±2.03 | 45.17±3.88 | 70.86±1.92 | 31.07±4.58 | 77.99±0.55 |
| | F1-Score | 67.58±1.16 | 44.67±0.97 | 58.32±3.08 | 40.21±2.01 | 46.10±4.70 | 65.13±1.72 | 35.67±4.69 | 69.44±0.00 |
| DO2HSC (Zhang et al., 2024) | AUC | 66.04±1.99 | 52.96±2.70 | 77.12±2.15 | 76.51±3.17 | 75.47±3.90 | 77.37±5.03 | 68.31±4.31 | 72.65±0.09 |
| | F1-Score | 61.14±2.67 | 50.00±2.72 | 70.87±2.73 | 75.61±2.73 | 73.28±2.44 | 79.80±0.50 | 66.63±3.04 | 68.67±0.09 |
| SDM-ATI | AUC | 94.93±0.03 | 89.19±0.17 | 85.71±11.37 | 88.50±3.22 | 62.92±0.62 | 86.53±0.00 | 90.52±2.93 | 95.05±1.48 |
| | F1-Score | 91.18±0.00 | 94.89±0.89 | 82.41±3.07 | 84.54±3.57 | 66.76±0.28 | 75.58±0.59 | 89.94±0.89 | 80.00±0.00 |
| SDM-ATII | AUC | 95.28±0.11 | 95.15±0.06 | 86.59±6.56 | 98.00±0.01 | 91.76±1.06 | 87.93±0.25 | 77.15±35.32 | 94.92±0.81 |
| | F1-Score | 88.47±0.21 | 86.67±0.00 | 79.53±4.85 | 91.75±0.00 | 86.29±0.58 | 81.25±0.00 | 73.96±31.63 | 91.11±1.11 |
| SDM-NAT | AUC | **95.91±2.55** | **98.25±0.02** | **90.71±1.17** | **99.23±0.42** | **99.94±0.06** | **96.99±2.71** | **98.74±1.59** | **96.67±1.67** |
| | F1-Score | **92.42±0.62** | **94.44±0.00** | **83.68±1.04** | **97.59±1.29** | **99.75±0.25** | **91.33±4.92** | **91.19±0.89** | **92.31±0.65** |

*Table 2.* Overall performance analysis for the three variants of SDM on One-Class GLAD (Tables 1 and 6).

| Metric | | SDM-ATI | SDM-ATII | SDM-NAT |
|---|---|---|---|---|
| AUC | Min std | 5/14 | 4/14 | **10**/14 |
| | std≤5% | 10/14 | 11/14 | **14**/14 |
| | Best Result | 2/14 | 2/14 | **14**/14 |
| F1-Score | Min std | 7/14 | 4/14 | **9**/14 |
| | std≤5% | 13/14 | 12/14 | **14**/14 |
| | Best Result | 2/14 | 1/14 | **14**/14 |

normal intervals with different gaps, where the learned decision boundaries enclose the normal training data tightly. The simulation results in Figure 3 strongly support our assumption and motivation.

### 3.3. Experiment on One-Class GLAD

We compare our SDM variants (SDM-ATI, SDM-ATII, and SDM-NAT) with state-of-the-art GLAD approaches under the one-class classification setting, where the evaluations cover various types of graph-structured data, including molecule, biology, and social network data. Table 1 and Table 6 (in **Appendix D**) present the experimental results in terms of AUC and F1-Score. We can observe that the three variants of SDM demonstrate consistent improvements over both graph kernel methods and GNN-based GLAD methods.

For instance, in the first class of PROTEINS, SDM-NAT achieves 95.91% AUC and 92.42% F1-Score, which significantly outperforms the latest baselines such as SIGNET (69.51% AUC, 64.39% F1-Score), MUSE (73.17% AUC, 67.58% F1-Score), and DO2HSC (66.04% AUC, 61.14% F1-Score). Particularly, we identify a "performance flip" phenomenon, where the performance of different classes in a dataset may have significant differences in approaches, such as most graph kernels, MUSE (on DD), and DO2HSC (on PROTEINS). It is evident that the "performance flip" phenomenon is largely absent in SDM variants, as they maintain robust and competitive performance across different classes. In terms of the stability of the proposed three SDM variants, we summarize a performance overview (covering Tables 1 and 6) in Table 2, where the results indicate that SDM-NAT achieved the best performance in most cases and consistently maintains the lowest standard deviation (under 5%) for both AUC and F1-Score. This highlights the strong stability and effectiveness of SDM-NAT compared to the other two adversarial variants.

### 3.4. Experiment on Large-scale Imbalanced GLAD

We evaluate the performance on four large-scale imbalanced datasets, including SW-620, MOLT-4, PC-3, and MCF-7, where the rare "active" status in anti-cancer molecules is

*Table 3.* Average AUCs and F1-Scores with standard deviation (10 trials) on large-scale imbalanced graph datasets. The best results are marked in **bold**, and "N/A" denotes that the result is not available.

| Method | SW-620 | | MOLT-4 | | PC-3 | | MCF-7 | |
|---|---|---|---|---|---|---|---|---|
| Supervised | AUC | F1-Score | AUC | F1-Score | AUC | F1-Score | AUC | F1-Score |
| GCN (Kipf & Welling, 2017) | 74.90±0.74 | 52.00±0.87 | 72.55±0.52 | 50.53±1.03 | 75.36±2.13 | 49.99±1.63 | 72.70±1.05 | 48.76±0.59 |
| GIN (Xu et al., 2019) | 78.61±2.85 | 58.47±5.43 | 75.86±1.60 | 55.43±6.52 | 78.44±1.67 | 59.07±4.15 | 69.54±1.15 | 58.27±3.20 |
| SOPOOL (Wang & Ji, 2020) | 75.51±5.06 | 58.11±2.86 | 75.11±0.97 | 56.20±3.64 | 69.37±1.53 | 57.80±3.74 | 75.64±2.17 | 56.82±3.57 |
| RWGNN (Nikolentzos & Vazirgiannis, 2020) | 73.37±0.36 | 51.31±1.62 | 71.30±1.23 | 50.67±2.64 | 76.27±0.86 | 50.44±2.76 | 70.47±1.26 | 51.65±3.46 |
| iGAD (Zhang et al., 2022a) | 85.82±0.69 | 63.68±1.56 | 83.59±1.07 | 63.30±1.17 | 86.04±1.14 | 63.50±0.73 | 83.22±0.64 | 64.70±2.58 |
| Unsupervised | AUC | F1-Score | AUC | F1-Score | AUC | F1-Score | AUC | F1-Score |
| GLocalKD (Ma et al., 2022) | 64.14±0.92 | 60.73±0.03 | 63.43±1.26 | 60.73±0.03 | 66.08±0.04 | 43.13±0.14 | 61.43±1.26 | 45.00±0.17 |
| OCGTL (Qiu et al., 2022) | 67.69±0.02 | 27.01±0.90 | 57.42±2.38 | 53.38±0.64 | 68.42±1.73 | 27.03±0.42 | 64.92±1.92 | 34.81±1.70 |
| SIGNET (Liu et al., 2023a) | 39.32±0.77 | 75.40±0.19 | 44.28±0.33 | 70.28±0.16 | 40.56±3.05 | 76.17±0.31 | 40.22±0.55 | 68.30±0.42 |
| MUSE (Kim et al., 2024) | N/A | N/A | N/A | N/A | 49.18±2.42 | 76.60±0.71 | 48.78±2.01 | 68.87±0.99 |
| DO2HSC (Zhang et al., 2024) | 43.12±0.70 | 33.65±0.66 | 51.51±2.39 | 42.30±1.34 | 52.25±3.18 | 35.66±1.26 | 53.08±2.38 | 43.73±1.32 |
| SDM-ATI | 90.19±8.94 | 87.40±0.21 | 90.25±7.57 | 83.59±0.24 | 91.59±6.73 | 86.82±2.12 | 81.62±8.18 | 80.61±0.04 |
| SDM-ATII | 92.91±5.48 | 91.61±0.00 | **97.05±2.39** | 83.76±0.07 | 94.30±0.63 | 87.47±1.13 | 88.40±0.13 | 82.34±0.12 |
| SDM-NAT | **94.26±2.86** | **93.16±1.99** | 94.20±4.79 | **90.36±2.39** | **97.09±1.78** | **94.17±1.70** | **94.71±2.13** | **88.98±1.80** |

treated as anomalies. Note that we exclude graph kernel methods for comparison due to their scalability constraints. Instead, we compare our SDM variants with the latest GNN-based approaches, including both unsupervised and supervised ones. Table 3 presents the comparison results, where we observe that supervised methods such as DCGNN and iGAD, significantly outperform unsupervised baselines like OCGTL and GLocalKD. This demonstrates that supervised methods can still leverage the limited labeled anomalies in large-scale imbalanced scenarios to facilitate anomaly detection. Nevertheless, supervised methods do not generalize well when the test data are not drawn from the same distribution as the training data, especially when the amount of available anomalous graphs is limited. Conversely, the proposed SDM variants, such as SDM-ATII and SDM-NAT, consistently surpass supervised baselines such as DGCNN and iGAD by more than 10% on most datasets. This can be attributed to the capability of SDM to generate high-quality pseudo-anomalous graphs for refining a more robust decision boundary without extensive reliance on labeled data. Additionally, SDM-NAT demonstrates better stability and accuracy compared to SDM-ATI, as it avoids the instability of GAN-based max-min optimization and explicitly defines an objective of generating high-quality anomalous graphs.

### 3.5. Experiment on Multi-Class GLAD

We conduct a multi-class graph-level anomaly detection experiment on ENZYMES to further demonstrate the effectiveness of our methods. In this experiment, multiple classes are regarded as anomalies. Specifically, we set the class $\{0, 1, 2, 3\}$ as the normal classes and $\{4, 5\}$ as the anomalous classes. Figure 4 shows the experimental results of our methods against several state-of-the-art GNN-based GLAD methods. We can observe that the proposed three methods significantly outperform all the baselines by a large margin (more than 20%). This demonstrates the feasibil-

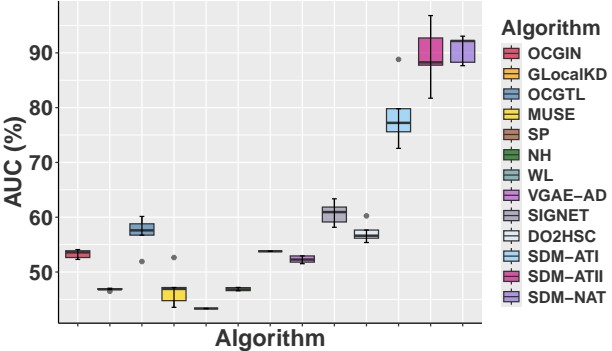

*Figure 4.* Experimental results of multi-class graph-level anomaly detection task on ENZYMES.

ity and potential of the proposed methods in dealing with multi-class GLAD scenarios. Moreover, SDM-NAT and SDM-ATII demonstrate better performance than SDM-ATI, and SDM-NAT exhibits more stability than SDM-ATII, as it has fewer performance fluctuations.

### 3.6. Analysis of Discriminative Scores

Figure 5 shows the discriminative scores of SDM-ATI, SDM-ATII, and SDM-NAT on MUTAG (class 1). The top row represents the discriminative scores in the training stage, while the bottom row corresponds to the testing stage. It is evident that during the training phase, the proposed three models effectively differentiate between the generated anomalous data and normal data, and this distinction carries over to the testing phase. These findings validate that the classifier trained using high-quality generated anomalous graphs can identify outstanding decision boundaries and exhibit excellent generalization capabilities during testing. Particularly, despite observing score overlap between the generated anomaly and normal graphs during the train-

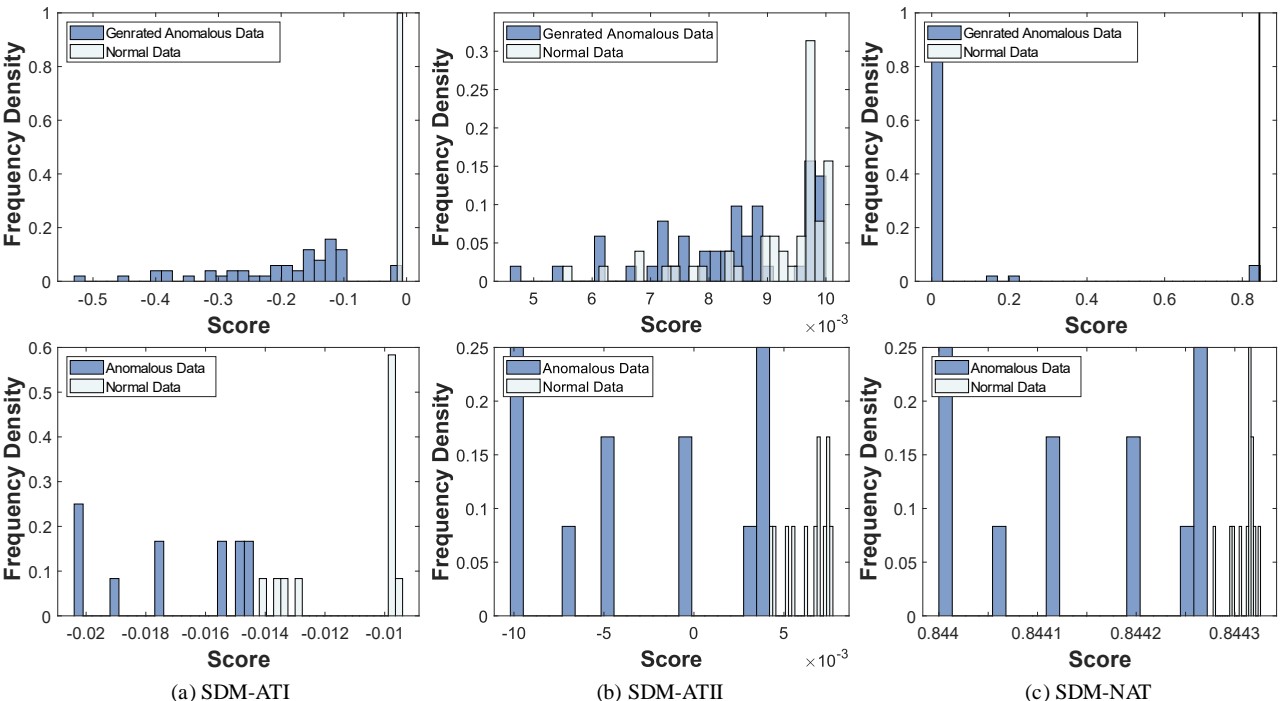

*Figure 5.* The discriminative scores of SDM-ATI, SDM-ATII, and SDM-NAT on MUTAG (Class 1), where the top and bottom rows represent the result of the training stage and testing stage, respectively. Note that the x-axis and y-axis indicate the output scores and the number density of data samples within a certain interval.

ing phase of SDM-ATII, significant differentiation is still achieved during the testing phase. This can be attributed to our objective of training a powerful classifier by generating high-quality anomaly graphs that closely resemble normal graphs. Although the classifier may not separate these anomalies adequately during training, which may be due to the over-idealization of the generated anomaly data, the learned decision boundaries are sufficiently effective in distinguishing the anomalies during the test phase.

### 3.7. More Experimental Analysis

Due to the page limitation, we provide more experimental analysis in our appendix, including the experimental results on more datasets (**Appendix** D), visualization analysis(**Appendix** E), parameter analysis (**Appendix** F), and ablation study on generator backbone (**Appendix** G), etc.

## 4. Conclusion

In this paper, we proposed a self-discriminative modeling (SDM) framework for graph-level anomaly detection. The key idea is to generate pseudo-anomalous graphs that interpolate between normal graphs and (real) anomalous graphs, though real anomalies are not presented in the training stage. We provide three variants of SDM, namely, SDM-ATI, SDM-ATII, and SDM-NAT. Particularly, SDM-NAT has

much higher learning stability and detection accuracy than the other two methods. The comprehensive experiments on various graph benchmark datasets, including molecular, biological, social network, and large-scale imbalanced molecular datasets, demonstrate the effectiveness of our methods compared to state-of-the-art graph-level anomaly detection methods. Surprisingly, although our methods are unsupervised learning, they outperformed a few strong baselines of semi-supervised learning methods. One limitation of our work is that we have not considered any real anomalous graphs in the training stage, though they may be available in some scenarios.

## Acknowledgements

This work was supported by the National Natural Science Foundation of China under Grant No.62376236 and the General Program of Natural Science Foundation of Guangdong Province under Grant No.2024A1515011771.

## Impact Statement

This paper presents work whose goal is to advance the field of Machine Learning. There are many potential societal consequences of our work, none of which we feel must be specifically highlighted here.

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

# A. Related Works

## A.1. Graph-level Anomaly Detection

Graph-level anomaly detection (Ma et al., 2021; Liu et al., 2024; Cai et al., 2024b;d) typically refers to the identification of graphs that are significantly different from others, *e.g.*, the active state of molecules or abnormal relational structure in social networks, which exhibit significant differences compared to normal graphs. Different from the anomaly detection tasks on node-level and edge-level (Ding et al., 2021; Huang et al., 2022; Duan et al., 2023; Qiao & Pang, 2023; Qiao et al., 2024b; Pan et al., 2025), graph-level anomaly detection demands the consideration of relationships between different graphs measured from the perspective of the entire graph. Graph kernels (GK), *e.g.*, Short-Path kernel (SP) (Borgwardt & Kriegel, 2005), Weisfeiler-Lehman kernel (WL) (Shervashidze et al., 2011), Random walk kernel (RW) (Vishwanathan et al., 2010), etc, are generally used to measure the similarity between pair-wise graphs (Akoglu et al., 2015). They can be utilized to achieve graph-level anomaly detection by combining with classical AD methods such as one-class support vector machine (OCSVM) (Schölkopf et al., 1999).

In recent years, graph neural networks (GNNs) (Zhou et al., 2020; Wu et al., 2020; 2023; Liu et al., 2023b; Li et al., 2023; Cai et al., 2024a; Wan et al., 2024; Cai et al., 2024c; Tu et al., 2025) have become a paradigm for learning graph representations. Many outstanding works take advantage of the graph learning capability of GNNs, such as GCN (Kipf & Welling, 2017; Wan et al., 2025) and GIN (Xu et al., 2019), to facilitate graph-level anomaly detection. For instance, Zhao & Akoglu (2021) combined GIN with DeepSVDD (Ruff et al., 2018) to construct an end-to-end graph anomaly detection framework. Qiu et al. (2022) leveraged graph transformation learning (Qiu et al., 2021) to alleviate the performance flip issue in graph anomaly detection. Ma et al. (2022) proposed to learn richer normal patterns of graphs from both local and global perspectives through the knowledge distillation within graph and node representations. Zhang et al. (2022b) investigated the problem of anomaly detection on imbalanced graphs and addressed this issue by introducing a supervised dual-discriminative framework with an imbalanced-specified loss function based on point mutual information. Compared with them, our methods do not require any assumption on the shape of the embedding distribution (*e.g.*, hypersphere (Zhao & Akoglu, 2021; Qiu et al., 2022; Zhang et al., 2024)) or explicitly defined anomaly scores (Ma et al., 2022; Liu et al., 2023a; Dong et al., 2024). Besides, our method is fully unsupervised but is able to generate high-quality pseudo-anomalous graphs for training the anomaly detector (classifier), which is more cost-efficient than supervised methods (Zhang et al., 2022b).

## A.2. GAN-based Anomaly Detection

As an emerging branch of deep learning techniques, generative adversarial network (GAN) (Goodfellow et al., 2014; Wang et al., 2021; Cai et al., 2024e) has been widely studied and utilized in anomaly detection, particularly in the vision area (Schlegl et al., 2019; Cai & Fan, 2022; Mou et al., 2023). For example, Schlegl et al. (2017) proposed AnoGAN, which applied GAN and unsupervised training to learn a manifold for normality, thereby identifying the anomalies in image data. Han et al. (2021) proposed a multiple GAN ensemble framework for anomaly detection, which better models normal data distribution through the interaction between multiple groups of generators and discriminators. Zhang et al. (2022b) leveraged self-supervised learning with a deep adversarial framework to capture the marginal distribution of normal image data for detecting anomalies. Nevertheless, although some recent works (Chen et al., 2020; Zheng et al., 2021; Xiao et al., 2023) explored the feasibility of GAN in node-level AD tasks, the studies and efforts on graph-level AD tasks that require taking both attributes and edges into account are still limited. To this end, our work endeavors to bridge this research gap.

We here discuss the connection between the proposed three variants of SDM (refer to Section 2) with the GAN-based approaches. First, SDM-ATI shares the idea of adversarial training in GAN, but they differ in several details. For example, the generator of SDM-ATI simultaneously produces anomalous nodes and edges for an anomalous graph, which differs from the others that focus solely on nodes or images. Besides, most GAN-based approaches adopt two-stage frameworks and define a specific anomaly score, *e.g.*, mapping loss function (Schlegl et al., 2017; 2019), reconstruction error (Wang et al., 2018; Lee et al., 2022) for detecting anomalies, while SDM-ATI is designed as a joint training framework and relies on the output of the discriminator. Second, SDM-ATII learns to generate high-quality anomalous samples from normal graphs, whereas other GAN-based approaches typically generate from random noise. More importantly, the variational inference (Kipf & Welling, 2016) involved in SDM-ATII guarantees the diversity of generated anomalous graphs. Third, we further proposed another variant, *i.e.*, SDM-NAT, to address the instability problem during training and generate high-quality anomalous graphs in a non-adversarial manner.

## B. Dataset

Table 4 illustrates the detailed information of each graph dataset used in our experiment, including the number of graphs, average nodes, average edges, node classes, graph classes, and data type. We also show the imbalance ratio of four large-scale graph datasets in Table 5.

*Table 4.* Detailed information of the graph benchmark datasets.

| Dataset Name | # Graphs | # Average $[V]$ | # Average $[E]$ | # Node Classes | # Graph Classes | Data Types |
|---|---|---|---|---|---|---|
| Small and moderate-scale datasets | | | | | | |
| PROTEINS | 1,113 | 39.06 | 72.82 | 3 | 2 | Biology |
| DD | 1,178 | 284.32 | 715.66 | 82 | 2 | Biology |
| ENZYMES | 600 | 32.63 | 62.14 | 3 | 6 | Biology |
| IMDB-BINARY | 1,000 | 19.77 | 96.53 | – | 2 | Social networks |
| ER_MD | 446 | 21.33 | 234.85 | 10 | 2 | Molecule |
| MUTAG | 188 | 17.93 | 19.79 | 7 | 2 | Molecule |
| AIDS | 2,000 | 15.69 | 16.20 | 38 | 2 | Molecule |
| COX2 | 467 | 41.22 | 43.45 | 8 | 2 | Molecule |
| Imbalanced large-scale datasets | | | | | | |
| SW-620 | 40,532 | 26.06 | 28.09 | 65 | 2 | Molecule |
| MOLT-4 | 39,765 | 26.10 | 28.14 | 64 | 2 | Molecule |
| PC-3 | 27,509 | 26.36 | 28.49 | 45 | 2 | Molecule |
| MCF-7 | 27,770 | 26.40 | 28.53 | 46 | 2 | Molecule |

*Table 5.* The imbalance ratio of large-scale graph benchmark datasets.

| Datasets | Class | # Graphs | Imbalance Ratio |
|---|---|---|---|
| SW-620 | Normal | 38,122 | 5.95% |
| | Anomalous | 2,410 | |
| MCF-7 | Normal | 25,476 | 8.26% |
| | Anomalous | 2,294 | |
| PC-3 | Normal | 25,941 | 9.34% |
| | Anomalous | 1,568 | |
| MOLT-4 | Normal | 36,625 | 7.90% |
| | Anomalous | 3,140 | |

## C. Detailed Experimental Settings

We consider three types of experiments in this paper to evaluate anomaly detection performance. (1) The first experiment focuses on the one-class classification task, where we treat each class of a dataset as the normal class and assess the anomaly detection performance for each class individually. (2) The second experiment involves anomaly detection on large-scale imbalanced graph datasets, where the class with a small number of samples is designated as the anomaly. (3) Moreover, we evaluate anomaly detection in multi-class scenarios, where the anomalies may come from multiple classes. Here, we describe the detailed experimental settings as follows:

- **Data Split:** For small and moderate-scale datasets, we allocate 80% of the data from the normal class for training, and subsequently construct the testing data by combining the remaining normal data with an equal or smaller number of anomalous data samples. For large-scale imbalanced datasets, we allocate 80% of the data in the normal class as the training set, and form the test set with the rest of the normal data and all the abnormal data.

- **Trade-off Parameters:** For SDM-ATI and SDM-ATII, we fix the clipping parameter $c$ of the adversarial loss at 0.01. Besides, SDM-ATII and SDM-NAT have two critical hyper-parameters, $\lambda$ and $\gamma$, in their loss functions, which control the contributions of the discrepancy loss and KL-divergence loss, respectively. We employ a grid search strategy, covering a range of $[0.001, 100]$, to explore the parameter space and determine their optimal values thoroughly. Furthermore,

*Table 6.* Average AUCs and F1-Scores with standard deviation (10 trials) on MUTAG, AIDS, and COX2. The best results are marked in **bold**.

| Method | Metric | MUTAG | | AIDS | | COX2 | |
|---|---|---|---|---|---|---|---|
| | | 0 | 1 | 0 | 1 | 0 | 1 |
| SP (Borgwardt & Kriegel, 2005) | AUC | 69.44±0.00 | 67.52±0.00 | 97.78±0.00 | 79.54±0.00 | 54.08±0.00 | 57.60±0.00 |
| | F1-Score | 58.33±0.00 | 60.00±0.00 | 95.00±0.00 | 76.56±0.00 | 49.32±0.00 | 45.00±0.00 |
| WL (Shervashidze et al., 2011) | AUC | 19.44±0.00 | 89.12±0.00 | 98.84±0.00 | 84.44±0.00 | 59.90±0.00 | 29.25±0.00 |
| | F1-Score | 16.67±0.00 | 76.00±0.00 | 93.75±0.00 | 83.55±0.00 | 60.27±0.00 | 35.00±0.00 |
| NH (Hido & Kashima, 2009) | AUC | 66.53±3.30 | 79.97±0.40 | 96.85±0.21 | 63.47±3.23 | 61.41±0.82 | 47.17±0.00 |
| | F1-Score | 56.67±6.24 | 76.00±0.00 | 97.50±0.00 | 59.25±1.35 | 56.44±1.03 | 50.00±3.16 |
| RW (Vishwanathan et al., 2010) | AUC | 94.44±0.00 | 86.98±0.00 | 73.61±0.00 | 58.31±0.00 | 52.43±0.00 | 28.75±0.00 |
| | F1-Score | 83.33±0.00 | 80.00±0.00 | 66.25±0.00 | 57.50±0.00 | 47.95±0.00 | 30.00±0.00 |
| VGAE-AD (Kipf & Welling, 2016) | AUC | 80.00±7.07 | 73.30±5.40 | 56.59±1.59 | 96.74±0.53 | 59.28±1.55 | 73.33±1.48 |
| | F1-Score | 78.33±6.67 | 65.56±2.22 | 55.34±2.91 | 93.94±0.51 | 59.19±0.00 | 73.00±5.10 |
| OCGIN (Zhao & Akoglu, 2021) | AUC | 88.40±2.14 | 74.66±1.68 | 94.38±0.10 | 20.56±4.52 | 59.64±5.78 | 56.83±7.68 |
| | F1-Score | 61.54±0.00 | 62.95±0.00 | 88.12±0.62 | 29.22±5.16 | 47.95±0.00 | 52.38±0.00 |
| GLocalKD (Ma et al., 2022) | AUC | 84.03±0.00 | 90.59±0.61 | **100.00±0.00** | 94.45±5.85 | 51.42±0.66 | 65.79±0.98 |
| | F1-Score | 83.33±0.00 | 86.17±0.91 | **100.00±0.00** | 88.27±7.70 | 51.24±0.60 | 65.33±0.67 |
| OCGTL (Qiu et al., 2022) | AUC | 93.61±0.24 | 87.04±1.74 | 98.09±0.48 | 99.49±0.08 | 60.42±0.90 | 54.65±3.09 |
| | F1-Score | 38.46±12.87 | 80.00±0.00 | 97.50±0.00 | 97.25±0.57 | 55.62±5.24 | 52.38±4.26 |
| SIGNET (Liu et al., 2023a) | AUC | 79.40±3.69 | 86.13±3.52 | 88.28±1.69 | 96.15±0.89 | 47.25±0.00 | 64.50±0.00 |
| | F1-Score | 80.56±3.93 | 80.00±3.27 | 80.25±2.00 | 89.69±1.06 | 52.05±0.00 | 60.00±0.00 |
| MUSE (Kim et al., 2024) | AUC | 83.79±7.02 | 87.73±2.45 | 55.96±2.12 | 92.56±3.03 | 54.14±3.23 | 48.08±7.04 |
| | F1-Score | 75.20±6.14 | 73.07±4.11 | 52.88±1.26 | 85.91±4.39 | 52.14±3.49 | 48.50±9.50 |
| DO2HSC (Zhang et al., 2024) | AUC | 94.72±4.47 | 88.83±6.58 | 91.09±2.34 | 88.03±0.91 | 63.16±3.36 | 72.28±4.67 |
| | F1-Score | 89.17±7.50 | 86.80±6.21 | 82.92±3.12 | 83.50±0.56 | 58.36±2.95 | 69.00±4.90 |
| SDM-ATI | AUC | 95.83±0.00 | 73.44±34.56 | **100.00±0.00** | **100.00±0.00** | 71.46±9.15 | 79.45±5.82 |
| | F1-Score | 83.33±0.00 | 91.67±0.00 | **100.00±0.00** | **100.00±0.00** | 70.78±7.91 | 67.10±4.17 |
| SDM-ATII | AUC | 99.31±1.42 | 81.50±25.66 | **100.00±0.00** | **100.00±0.00** | 81.51±0.31 | 91.78±3.95 |
| | F1-Score | 99.13±1.74 | 80.27±23.47 | 99.37±0.63 | **100.00±0.00** | 74.05±0.95 | 84.79±3.38 |
| SDM-NAT | AUC | **100.00±0.00** | **99.36±0.35** | **100.00±0.00** | **100.00±0.00** | **96.66±0.00** | **97.05±1.62** |
| | F1-Score | **100.00±0.00** | **98.53±0.20** | **100.00±0.00** | **100.00±0.00** | **90.41±0.00** | **92.50±2.50** |

**Appendix** F contains the evaluation of the impact of variations in the values of $\lambda$ and $\gamma$ on the anomaly detection performance.

- **Baseline Settings:** We achieve anomaly detection for all graph kernels and InfoGraph by combining them with OCSVM (Schölkopf et al., 1999). For other baseline methods, *e.g.*, OCGIN, OCGTL, GLocalKD, iGAD, SIGNET, MUSE, and DO2HSC, we follow the settings in their papers and reproduce the experimental results with the officially released codes. To guarantee a fair comparison, we use the same network structure, *i.e.*, GIN (Xu et al., 2019), for each GNN-based baseline and SDM variants. Particularly, in the imbalanced graph anomaly detection task, we apply the cross-entropy loss function to the supervised GNN baselines, *e.g.*, GCN, DGCNN, GIN, SOPOOL, and RWGNN.

- **Training Details:** For small-scale graph datasets, the grid search strategy is utilized to find the optimal performance, where the coefficients ($\lambda$ and $\gamma$) vary in $\{0.1, 1, 10\}$, and the batch size varies in $\{4, 8, 16\}$, while we increase the batch size to 256 to accommodate the requirement of experiment on larger-scale datasets. We utilize RMSprop (Tieleman et al., 2012) as the optimizer for SDM-ATI and SDM-ATII, and Adam (Kingma & Ba, 2014) for SDM-NAT during training. Besides, we set the learning rate $\rho$ to 0.001 with the total training epochs to 300.

- **Implementation:** We leverage PyTorch Geometric (Fey & Lenssen, 2019) for implementation, and all experiments are executed on an NVIDIA Tesla A100 GPU with an AMD EPYC 7532 CPU.

# D. More Experimental Results

### D.1. Performance Comparison on More Datasets

Table 6 presents an additional evaluation of our SDM framework (SDM-ATI, SDM-ATII, and SDM-NAT) against various graph kernel methods and GNN-based GLAD methods on four benchmark datasets, including MUTAG, AIDS, and COX2. We can observe that our SDM variants consistently outperform baseline methods across all benchmark datasets. Particularly, SDM-NAT performance achieves 96.66% (AUC) and 90.41% (F1-Score) on the challenging dataset COX2 (class 0), which marks a more than 20% improvement over other state-of-the-art approaches such as OCGTL and DO2HSC. Notably, SDM-NAT exhibits markedly lower variance than adversarial strategies (SDM-ATI, SDM-ATII) and competing baselines, which highlights its robustness in decision boundary learning. Moreover, we notice that certain baselines show substantial performance drops when setting another class as the normal class, *e.g.*, the AUC decline of OCGIN on AIDS and SIGNET on COX2, which implies the challenges of handling heterogeneous distributions without strong assumptions. In contrast, the proposed SDM framework achieves consistent efficacy across graph datasets from diverse domains, demonstrating its broad applicability to real-world scenarios.

### D.2. Explanation of SDM's Advantages on Imbalance GLAD

To better understand the success of SDM, we provide a visual illustration in Figure 6 to explain why the SDM can outperform other supervised methods in the context of imbalanced scenarios (Kim et al., 2024; Qi et al., 2025). This figure demonstrates a binary classification scenario where the anomalous data used for training lies on the right side of the normal data, and the trained classifier successfully categorizes them with a red decision boundary. However, unknown anomalies may be located on the left side of the decision boundary (shown by the blue dashed line), where the binary classifier fails to detect them. This phenomenon is common in large-scale unbalanced anomaly detection due to limited supervised information, where the distribution of test data is not exactly the same as that of the training data.

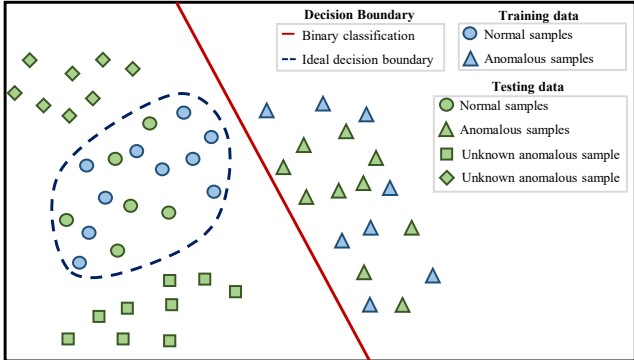

*Figure 6.* The illustration of the advantages of SDM. The **red** line denotes the decision boundary learned by the binary classification, whereas the **blue** dashed line denotes the ideal decision boundary.

# E. Visual Analysis of Learned Decision Boundary

In this section, we aim to intuitively demonstrate the effectiveness of the proposed methods. Specifically, we provide visualization results of learned embeddings using t-SNE (Van der Maaten & Hinton, 2008). We show the t-SNE visualizations (on MUTAG and AIDS) for the learned embeddings of several baseline methods and the three variants of SDM in Figures 7 and 8, enabling a more comprehensive assessment. These two figures show that the proposed SDM methods learn more explicit decision boundaries to distinguish anomalous samples compared to the visualization results of other baseline methods. Besides, these figures also offer compelling insights into the learned decision boundaries derived from the generated anomalous data. We can observe that the normal data from both training and testing stages approximately lie on the same manifold, which implies that the normality is well captured in our methods. Moreover, the real anomalous and generated pseudo-anomalous data are well separated into different regions from the normal data. This distinct separation serves as compelling evidence of the discriminative power implicit in our methods.

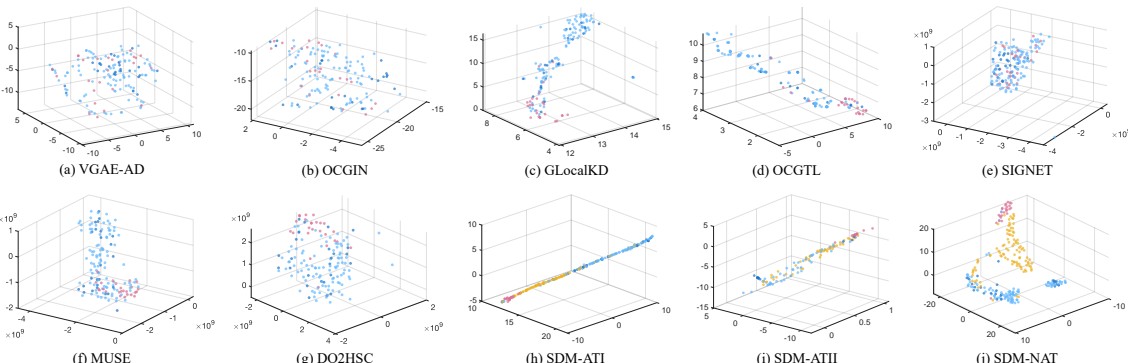

*Figure 7.* The t-SNE visualization on MUTAG (Class 1). Note that the points marked in **red**, **light blue**, and **dark blue** represent the real anomalous data, and the normal data in training and testing stages, while points marked in **yellow** denote the generated anomalous data in our methods.

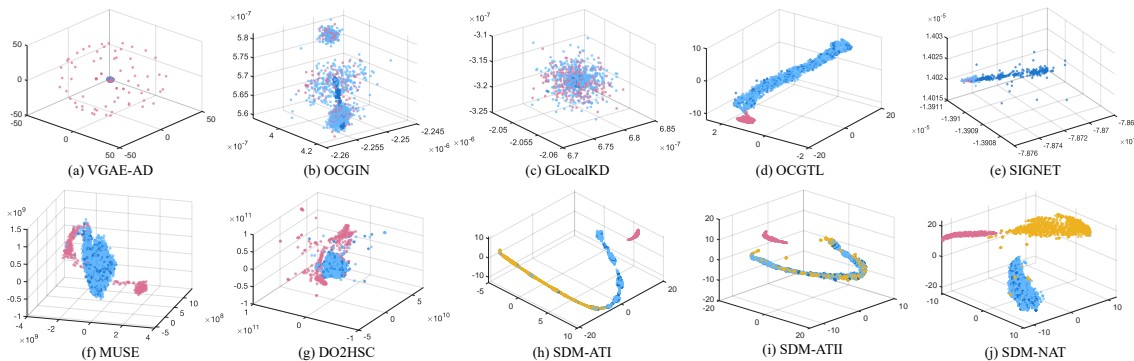

*Figure 8.* The t-SNE visualization on AIDS (Class 1). Note that the points marked in **red**, **light blue**, and **dark blue** represent the real anomalous data, and the normal data in training and testing stages, while points marked in **yellow** denote the generated anomalous data in our methods.

Nevertheless, it is worth noting that t-SNE would change the actual data distribution through dimensionality reduction. Therefore, the distribution of generated anomalies may not encompass the normal data in the visualization results. However, we can still observe that more discriminative latent representations are learned in SDM variants compared to other baselines. This observation demonstrates the strong discrimination of the trained classifier, which is attributed to the high-quality pseudo-anomalous graphs generated adaptively. These insightful visualizations offer a better understanding of the proposed methods, vividly illustrating their ability to learn effective decision boundaries and unveil intricate patterns and anomalies hidden within complex graph structures.

## F. Parameter Sensitivity Analysis

We investigate the impact of two main hyper-parameters, $\lambda$ and $\gamma$, in SDM-ATII and SDM-NAT on the anomaly detection performance. Note that SDM-ATI is not included in this analysis because its loss function does not have any hyper-parameters. Specifically, we set the value of $\lambda$ and $\gamma$ from a wide range, *i.e.*, $[1e^{-3}, 1e^2]$, to evaluate their impact on anomaly detection performance. Figures 9 and 10 show the experimental results on COX2 and ER_MD, where we have the following observations.

First, we find that a balanced trade-off of $\lambda$ and $\gamma$ is crucial to achieve ideal performance for SDM-ATII and SDM-NAT. As illustrated in these two figures, we can observe that either excessively large or small values of $\lambda$ and $\gamma$ typically yield sub-optimal results in various cases. Second, we can observe that both SDM-ATII and SDM-NAT exhibit relatively stable performance across a broad range of $[1e^{-3}, 1e^2]$ for $\lambda$ and $\gamma$ values, particularly on ER_MD (Class 0) where SDM-ATII and SDM-NAT show stable performance in terms of AUC and F1-Score. Additionally, from the experimental results on other

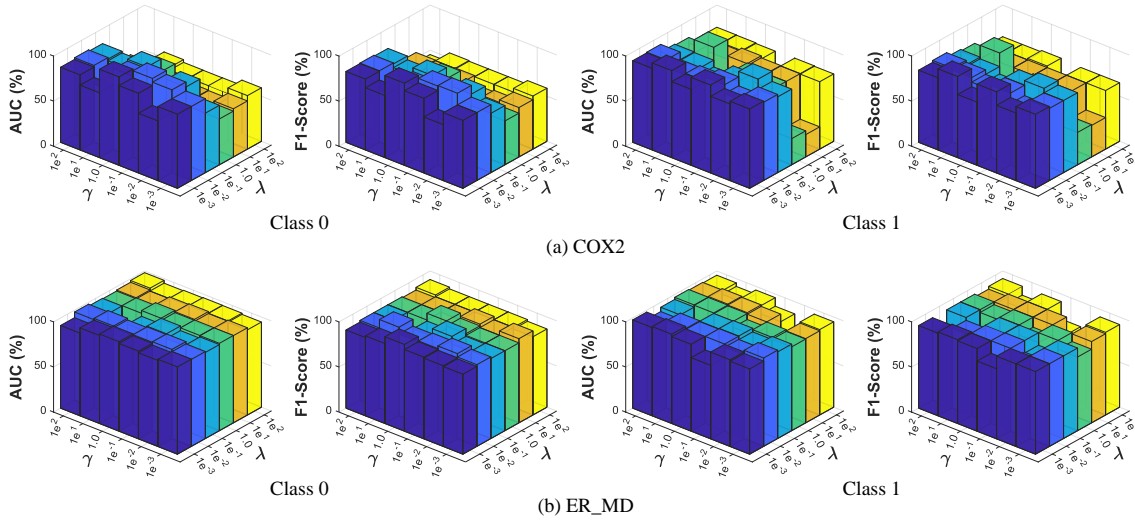

Figure 9. Parameter sensitivity analysis of SDM-ATII on COX2 and ER_MD. Note that the values of $\lambda$ and $\gamma$ changes in the range of $[1e^{-3}, 1e^2]$.

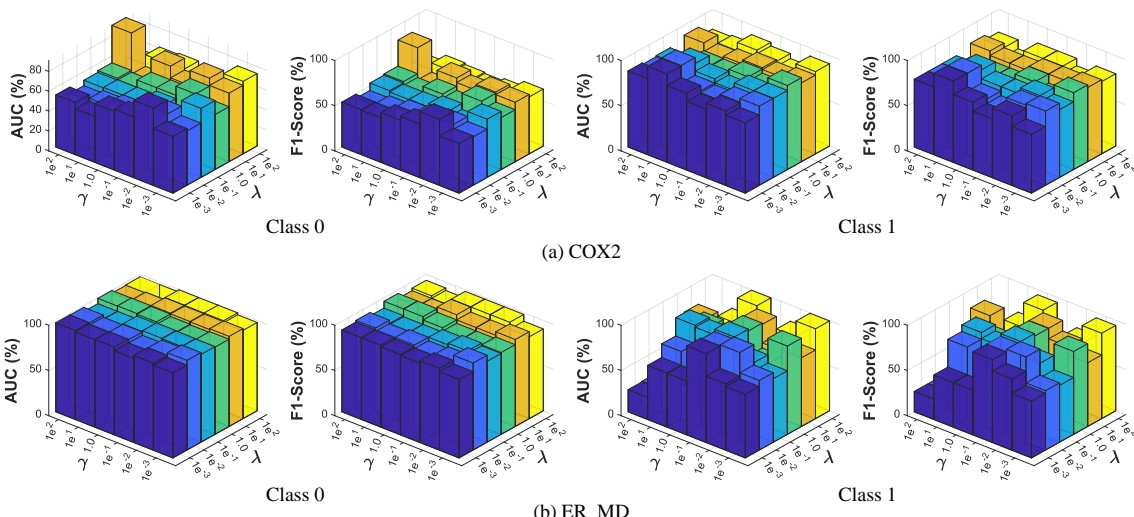

Figure 10. Parameter sensitivity analysis of SDM-NAT on COX2 and ER_MD. Note that the values of $\lambda$ and $\gamma$ change in the range of $[1e^{-3}, 1e^2]$.

cases, we can observe that the performance variation of both methods under varying $\lambda$ and $\gamma$ values also remains marginal within a relatively wide range. These observations fully demonstrate the stability and robustness of our methods.

## G. Ablation Study of Generator Backbone

We conduct an ablation study by thoroughly comparing VGAE-based and GIN-based generators to elucidate our rationale for leveraging VGAE as the preferred backbone for generators in SDM. It should be noted that the key difference between VGAE-based and GIN-based backbones lies in the incorporation of variational inference, which introduces stochasticity in generating anomalous graphs. Figure 11 shows a comprehensive performance comparison in terms of AUC across three graph datasets, including COX2, DD, and MUTAG.

From this figure, we can observe that the VGAE-based backbone consistently outperforms the GIN-based backbone by a substantial margin. This significant improvement can be attributed to the inherent disparities in their respective generation

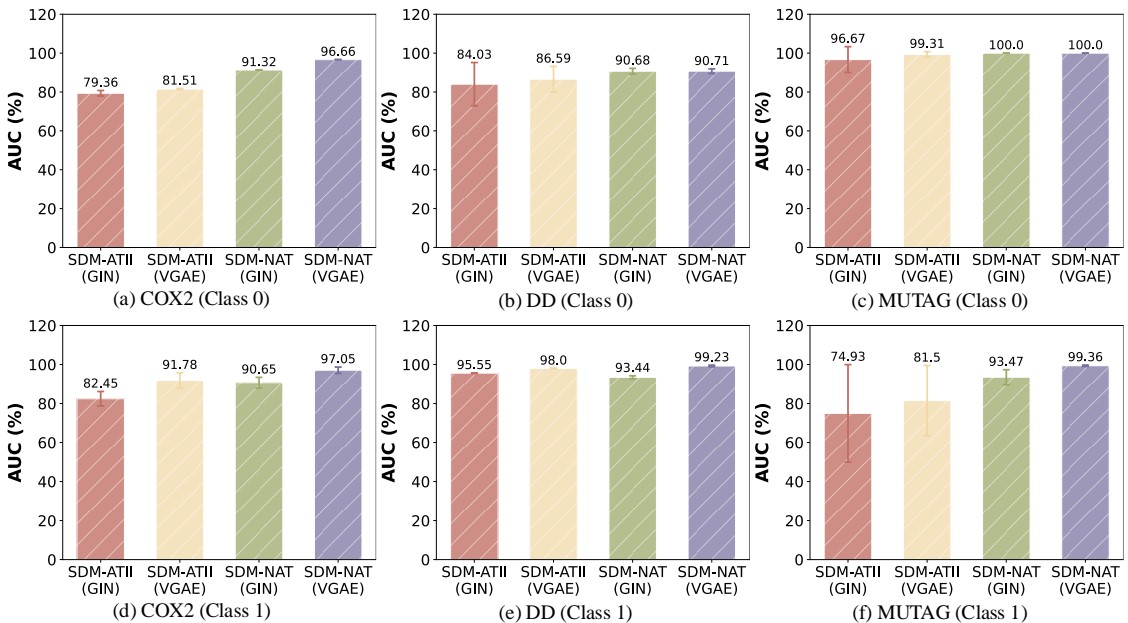

*Figure 11.* Performance comparison between VGAE-based and GIN-based generator.

processes. The GIN-based backbone generates graphs deterministically, while the VGAE-based generator incorporates stochasticity. In contrast to the GIN-based backbone, VGAE employs the reparameterization technique to learn a target distribution, allowing it to capture the data and underlying distribution. Consequently, the latent distribution of generated pseudo-anomalous data is more likely to reside in plausible regions rather than simply approximating the normal data. The experiment demonstrates the exceptional ability of the VGAE-based generator to produce high-quality pseudo-anomalous graphs, yielding superior performance in graph-level anomaly detection tasks. This also aligns with the motivation and expectation depicted in Figure 6, where the stochasticity in the generation process plays an important role in learning a good decision boundary.

## H. Robustness Analysis under Data Contamination

To evaluate the robustness of the proposed SDM under data contamination, we conduct an experiment by injecting anomalous data into the training set at varying contamination levels, which is defined as a certain percentage of the normal data (0%, 10%, 20%, and 30%, respectively). Table 7 shows the experimental results on MUTAG (class 0), where we can observe that the three SDM variants maintain stable performance under different contamination levels. In particular, the performance of SDM variants under a 10% contamination level still outperforms the SOTA baseline, such as DOH2SC (94.72% AUC w/o contamination). These results fully demonstrate the robustness and real-world applicability of our approach.

*Table 7.* Comparison of model performance at different contamination levels.

| Contamination Level | | 0% | 10% | 20% | 30% |
|---|---|---|---|---|---|
| SDM-ATI | AUC | **95.83(0.00)** | 94.77(0.79) | 93.85(0.04) | 94.23(1.23) |
| | F1-Score | **83.33(0.00)** | 82.61(0.46) | 80.24(0.00) | 81.27(1.96) |
| SDM-ATII | AUC | **99.31(1.42)** | 97.58(0.48) | 97.25(0.02) | 97.04(0.15) |
| | F1-Score | **99.13(1.74)** | 94.12(0.00) | 88.24(0.00) | 89.22(0.98) |
| SDM-NAT | AUC | **100.00(0.00)** | 97.67(0.73) | 96.44(0.21) | 95.71(2.17) |
| | F1-Score | **100.00(0.00)** | 92.16(1.96) | 88.24(0.00) | 87.70(3.39) |

# I. Detailed Algorithm

To facilitate the understanding of the training procedure of the proposed SDM methods, we provide the detailed algorithms of three SDM variants in Algorithms 1, 2, and 3.

---

**Algorithm 1** SDM-ATI

---

**Input:** Input graph set $\mathbb{G}$, number of GIN layers $K$, clipping parameter $c$, learning rate $\rho$, batch size $m$, total training epochs $\mathcal{T}$.

**Output:** The anomaly detection scores.

1: Initialize the network parameters $\phi, \omega$;
2: **for** $t \to \mathcal{T}$ **do**
3:     **for** each batch **G do**
4:         **Update Generator:**
5:         Unfreeze the the parameter $\phi$ of generator $\mathcal{G}_\phi$;
6:         Freeze the the parameter $\omega$ of discriminator $\mathcal{D}_\omega$;
7:         Sample random variable $\tilde{\mathbf{Z}}$ from latent Gaussian distribution $\mathbb{P}_{\tilde{\mathbf{Z}}} \sim \mathcal{N}(\mathbf{0}, \mathbf{1})$;
8:         Generate anomalous graph set $\tilde{\mathbf{G}}$ from generator $\mathcal{G}_\phi$ with the input $\tilde{\mathbf{Z}}$ via Eq. (6);
9:         Update the parameter $\phi$ of generator $\mathcal{G}_\phi$ by
        $\mathcal{G}_\phi \leftarrow \nabla[-\frac{1}{m} \sum_{i=1}^{m} \mathcal{D}_\omega(\mathcal{G}_\phi(\tilde{\mathbf{Z}}_i), \mathcal{T}(\mathcal{G}_\phi(\tilde{\mathbf{Z}}_i)\mathcal{G}_\phi(\tilde{\mathbf{Z}}_i)^\top))];$
        $\phi \leftarrow \phi - \rho \cdot \text{RMSProp}(\phi, \mathcal{G}_\phi);$
10:       **Update Discriminator:**
11:       Freeze the the parameter $\phi$ of generator $\mathcal{G}_\phi$;
12:       Unfreeze the the parameter $\omega$ of discriminator $\mathcal{D}_\omega$;
13:       Repeat steps 6 - 7;
14:       Update the parameter $\omega$ of generator $\mathcal{D}_\omega$ by
        $\mathcal{D}_\omega \leftarrow \nabla[-\frac{1}{m} \sum_{i=1}^{m} \mathcal{D}_\omega(\mathbf{X}_i, \mathbf{A}_i) + \frac{1}{m} \sum_{i=1}^{m} \mathcal{D}_\omega(\mathcal{G}_\phi(\tilde{\mathbf{Z}}_i), \mathcal{T}(\mathcal{G}_\phi(\tilde{\mathbf{Z}}_i)\mathcal{G}_\phi(\tilde{\mathbf{Z}}_i)^\top))];$
        $\omega \leftarrow \omega - \rho \cdot \text{RMSProp}(\omega, \mathcal{D}_\omega);$
        $\omega \leftarrow \text{clip}(\omega, -c, c);$
15:     **end for**
16: **end for**
17: Compute anomaly detection scores for test graphs via the trained discriminator $\mathcal{D}_\omega$;
18: **Return:** The anomaly detection scores.

---

---

**Algorithm 2** SDM-ATII

---

**Input:** Input graph set $\mathbb{G}$, number of GIN layers $K$, clipping parameter $c$, learning rate $\rho$, batch size $m$, total training epochs $\mathcal{T}$.

**Output:** The anomaly detection scores.

1: Initialize the network parameters $\phi, \omega$;
2: **for** $t \rightarrow \mathcal{T}$ **do**
3:     **for** each batch **G do**
4:         **Update Generator:**
5:         Unfreeze the the parameter $\phi$ of generator $\mathcal{G}_\phi$;
6:         Freeze the the parameter $\omega$ of discriminator $\mathcal{D}_\omega$;
7:         Extract graph-level representation with the input normal attributes $\mathbf{X}$ and adjacency matrix $\mathbf{A}$ via Eqs. (7) and (8);
8:         Generate anomalous anomalous graph set $\tilde{\mathbf{G}}$ from generator $\mathcal{G}_\phi$ with normal attributes $\mathbf{X}$ and adjacency matrix $\mathbf{A}$ via Eqs. (10), (11), (12), and (13);
9:         Update the parameter $\phi$ of generator $\mathcal{G}_\phi$ by
        $\mathcal{G}_\phi \leftarrow \nabla[-\frac{1}{m}\sum_{i=1}^{m}\mathcal{D}_\omega(\mathcal{G}_\phi(\mathbf{X}_i, \mathbf{A}_i) + \frac{\lambda}{m}\sum_{i=1}^{m}(\|\mathbf{X}_i - \tilde{\mathbf{X}}_i\|_F^2 - (\mathbf{A}_i\log(\tilde{\mathbf{A}}_i) + (1 - \mathbf{A}_i)\log(1 - \tilde{\mathbf{A}}_i))) + \gamma KL[q(\mathbf{Z}_\mathbb{G}|\mathbf{H}_\mathbb{G}, \mathbf{A})\|P(\mathbf{Z})]];$
        $\phi \leftarrow \phi - \rho \cdot \text{RMSProp}(\phi, \mathcal{G}_\phi);$
10:       **Update Discriminator:**
11:         Freeze the the parameter $\phi$ of generator $\mathcal{G}_\phi$;
12:         Unfreeze the the parameter $\omega$ of discriminator $\mathcal{D}_\omega$;
13:         Repeat steps 6 - 7;
14:         Update the parameter $\omega$ of generator $\mathcal{D}_\omega$ by
        $\mathcal{D}_\omega \leftarrow \nabla[-\frac{1}{m}\sum_{i=1}^{m}\mathcal{D}_\omega(\mathbf{X}_i, \mathbf{A}_i) + \frac{1}{m}\sum_{i=1}^{m}\mathcal{D}_\omega(\mathcal{G}_\phi(\mathbf{X}_i, \mathbf{A}_i))];$
        $\omega \leftarrow \omega - \rho \cdot \text{RMSProp}(\omega, \mathcal{D}_\omega);$
        $\omega \leftarrow \text{clip}(\omega, -c, c);$
15:     **end for**
16: **end for**
17: Compute anomaly detection scores for test graphs via the trained discriminator $\mathcal{D}_\omega$;
18: **Return:** The anomaly detection scores.

---

**Algorithm 3** SDM-NAT

---

**Input:** Input graph set $\mathbb{G}$, number of GIN layers $K$, learning rate $\rho$, batch size $m$, total training epochs $\mathcal{T}$.

**Output:** The anomaly detection scores.

1: Initialize the network parameters $\phi, \theta$ for anomalous generator $\mathcal{G}_\phi$ and classifier $f_\theta$;
2: **for** $t \rightarrow \mathcal{T}$ **do**
3:     **for** each batch **G do**
4:         Extract graph-level representation with the input normal attributes $\mathbf{X}$ and adjacency matrix $\mathbf{A}$ via Eqs. (7) and (8);
5:         Generate anomalous graph set $\tilde{\mathbb{G}}$ from generator $\mathcal{G}_\phi$ with normal attributes $\mathbf{X}$ and adjacency matrix $\mathbf{A}$ via Eqs. (10), (11), (12), and (13);
6:         Calculate the anomalous reconstruction loss via Eq. (14);
7:         Calculate the total loss via Eq. (16);
8:         Update the parameter $\phi$ and $\theta$ of anomalous generator $\mathcal{G}_\phi$ and classifier $f_\theta$ using backpropagation;
9:     **end for**
10: **end for**
11: Compute anomaly detection scores for test graphs via the trained classifier $f_\theta$;
12: **Return:** The anomaly detection scores.

---

