# OpenReview forum: "Self-Discriminative Modeling for Anomalous Graph Detection"
_ICML.cc/2025/Conference — ICML 2025 poster_

### Official Review · Reviewer_L2j6 · 2025-03-10

**Overall Recommendation:** 4

**Summary:**

The paper presents self-discriminative modeling method for graph-level anomaly detection. By generating pseudo-anomalous graphs to interpolate between normal and anomalous samples, the method constructs a reliable decision boundary solely based on normal data. The claims are well supported by corresponding theory and analyses, and comparison experiments on various graph benchmarks validates the effectiveness of the proposed method.

**Claims And Evidence:**

This paper asserts that an accurate decision boundary for normal data can be determined by generating anomalous data that closely resembles normal data. This claim is supported by both theoretical analysis and simulation experiments.

**Essential References Not Discussed:**

No other references are necessarily to be cited/discussed.

**Experimental Designs Or Analyses:**

The experiments in this paper are comprehensive and the experimental comparison is fair. The results are consistent to related analyses.

**Methods And Evaluation Criteria:**

The proposed methods are effective and novel, and the evaluations are justified.

**Other Comments Or Suggestions:**

Please refer to Weaknesses.

**Other Strengths And Weaknesses:**

- Strengths:
    1. This paper presents a clear and logical structure, making it easy to understand. The proposed approach appears promising for graph-level anomaly detection.
    2. The motivation is well-articulated and reinforced through both theoretical analysis and experimental validation.
    3. The study includes extensive comparative experiments, providing strong evidence of the method’s effectiveness.
- Weaknesses:
    1. How do the three proposed methods control the gap between pseudo-anomalous graphs and normal graphs without prior knowledge of real anomalous data? Does this introduce difficulties for the classifier in distinguishing them?
    2. In Figure 3, there are significant overlaps between pseudo-anomalous and normal data. If these overlapping pseudo data points were removed, would the proposed method perform better?
    3. Could the authors provide visualizations of the generated anomalous and normal data? This would greatly enhance the credibility of the approach.

**Questions For Authors:**

Please refer to the Weaknesses.

**Relation To Broader Scientific Literature:**

N/A

**Theoretical Claims:**

The theoretical claim is that the proposed model can effectively distinguish normal graphs from the generated pseudo-anomalous graphs, which serve as intermediates between normal and real anomalies. Additionally, the method for generating pseudo-anomalous graphs is well justified.

---

> ### Author Rebuttal · Authors · 2025-03-31
>
> **To W1:** Thank you for the insightful question. The adversarial training of SDM-ATI and SDM-ATII utilizes a generator to produce pseudo-anomalous graphs, with a discriminator balancing this via an adversarial loss (Eq. 9). Therefore, the numbers of training epochs for the generator and discriminator inherently control the gap between normal and pseudo-anomalous graphs. In practice, we employed a balanced setting of training epochs for both the generator and discriminator, and achieved superior performance. As visual evidence illustrated by the t-SNE plots (Figure 7), although generated anomalies stay close to normal data, they remain sufficiently distinct. SDM-NAT controls the gap through $\lambda$, a larger $\lambda$ implies a larger similarity between the generated pseudo anomalous graphs and the normal graphs, but we can observe that the performance is robust across a wide range of $\lambda$ (see Figure 9).
>
> We want to clarify that this setting would not introduce difficulties in training the classifier, as the adversarial dynamic and the joint learning in our methods can adaptively balance the discrimination capability of the classifier and the generation quality of the generator. The performance comparison (Tables 1, 3, and 6) across multiple datasets is good evidence to support this claim, and the embedding separation (e.g., Figure 7) also confirms the discrimination capability of the classifier. Besides, we further provided statistical analysis to demonstrate that pseudo-anomalous graphs are sufficiently different from normal graphs, so that they can truly benefit decision boundary learning (see response to Reviewer 1ZeV's W1).
>
> **To W2:** Thank you for your valuable concern. We tested the performance by removing the overlapping generated pseudo graph anomalies but gained only a slight improvement (usually $\leq 1\\%$, sometimes even drops) in performance. Actually, our method doesn't require explicitly distinguishing between normal and pseudo anomaly graphs. Our results (e.g., Figure 6) show that despite overlaps observed between normal and pseudo anomalies during training, SDM variants still maintain superior generalization performance in the test phase and achieve robust separation of normal and anomalous graphs.
>
> **To W3:** We appreciate the suggestion. Actually, we have included relevant visualization results in Appendix E.2. For example, in Figure 7, we not only visualize the normal and real anomalous graphs for our methods (as in other baselines), but also visualize the generated pseudo anomalous graphs for comparison. The data points marked in yellow denote the generated pseudo-anomalous graphs in our methods. We observe that SDM effectively separates normal and anomalous graphs, with pseudo-anomalous graphs interpolated between normal and real anomalous graphs (refer to Figure 7 (h), (i), and (j)). This serves as good evidence to support the strengths of those pseudo-anomalous graphs in learning a more robust decision boundary compared to other baseline methods.
>
> **Again, we thank the reviewer for recognizing our work, and we hope that our responses can solve your concerns.**

---

> > ### Comment · Reviewer_L2j6 · 2025-04-03
> >
> > Thank you for the detailed responses. The explanations and additional results have solved my previous questions. I also briefly reviewed the authors' responses to other reviewers. Overall, the paper is well-executed and makes a meaningful contribution to the anomaly detection community. I am therefore inclined to raise my rating.

---

> > > ### Author Response · Authors · 2025-04-03
> > >
> > > Dear Reviewer L2j6:
> > >
> > > Thank you for taking the time to review our responses and additional results. We are pleased that our clarifications help to address your concerns, and appreciate your recognition of our work.
> > >
> > > Best regards,
> > >
> > > Authors

---

### Official Review · Reviewer_1ZeV · 2025-03-11

**Overall Recommendation:** 4

**Summary:**

This paper proposes a novel GLAD framework named Self-Discriminative Modeling (SDM). The key idea of SDM is to generate pseudo-anomalous graphs from normal graphs and train a classifier/discriminator to distinguish them. The generative model and discriminative model are jointly trained to learn a more robust decision boundary. Moreover, the authors further introduce a non-adversarial variant for SDM. Experiments on 12 benchmark datasets demonstrate the SDM variants achieve superior performance compared to state-of-the-art GLAD baselines.

**Claims And Evidence:**

The main claim of this paper is that SDM can effectively detect anomalous graphs by leveraging the generated pseudo-anomalous graphs to refine a more reliable decision boundary. This claim is well-supported by experimental results.

**Essential References Not Discussed:**

No other references are necessarily to be cited/discussed. The authors have comprehensively reviewed the latest GLAD literature. They have broadly discussed the connection of the proposed framework with existing GLAD methods and GAN-based methods.

**Experimental Designs Or Analyses:**

I have checked the soundness/validity of the experimental designs and analysis. The experiments in this paper are comprehensive, covering diverse real-world GLAD scenarios. The data split strategy is consistent for each competitive method, the comparison is fair, and the experimental analysis is thorough and in-depth.

**Methods And Evaluation Criteria:**

The propose SDM methods make sense for the GLAD problem in real-world scenarios. The benchmark datasets are selected from diverse real-world domains, including molecular, biological, and social graphs with both balanced and imbalanced settings. The evaluation metrics (AUC, F1-Score) are standard and appropriate to evaluate anomaly detection performance.

**Other Comments Or Suggestions:**

1.	Please harmonize the formatting of “i.e.” and “e.g.”, such as unifying them as italics or not.
2.	Authors should further list in Table 4 the ratio of samples in each category.

**Other Strengths And Weaknesses:**

**Strengths:**
1. This paper is well-written. The idea of generating pseudo-anomalous graphs as the anomaly proxy for robust decision boundary learning is innovative and meaningful in addressing several key challenges in existing GLAD approaches.
2. The motivation of the proposed SDM framework is clearly illustrated, and it is well supported by the theoretical analysis (Sec. 2.1) and simulation experiment (Sec. 3.2). Moreover, three variants of SDM are designed to focus on different challenges.
3. The comparison experiments are extensive and demonstrate the effectiveness of SDM over state-of-the-art GLAD baselines in various scenarios such as one-class, multi-class, and large-scale imbalanced GLAD.

**Weaknesses:**
1. As the pseudo graph anomaly generation is the core of the proposed SDM, the major concern is whether the pseudo-anomalous graphs are sufficiently different from normal graphs so that they can truly benefit decision boundary learning.
2. The proposed framework generates pseudo-anomalous graphs by perturbing normal graphs. However, how does the method ensure that it captures diverse potential anomalies rather than a narrow subset?
3. In Fig. 6, it can be observed that the pseudo graph anomalies generated on SDM-ATII overlap significantly with the distribution of normal graphs (top row). The authors should further discuss this observation and its impact on model training.
4. This paper did not involve the robustness analysis of SDM under data contamination situations, which is common in real-world scenarios.

**Questions For Authors:**

Please refer to the Weaknesses part above.

**Relation To Broader Scientific Literature:**

Compared to recent GLAD approaches (e.g., [1, 2]) which imposes strict assumptions (e.g., hypersphere) on the latent space, SDM offers a more flexible solution by refining the decision boundaries through pseudo anomaly generation. Moreover, the unsupervised scheme of SDM also makes it more practical compared to supervised approaches [3].

[1] Raising the bar in graph-level anomaly detection, IJCAI, 2022.

[2] Deep orthogonal hypersphere compression for anomaly detection. ICLR, 2024.

[3] Dual-discriminative graph neural network for imbalanced graph-level anomaly detection, NeurIPS, 2022.

**Theoretical Claims:**

The analysis in Sec. 2.1 theoretically describes the motivation of generating pseudo graph anomalies to refine a more robust decision boundary, and the simulation results in Sec. 3.2 well support this claim.

---

> ### Author Rebuttal · Authors · 2025-03-31
>
> **We thank the reviewer for recognizing our work. Below are our responses to your concerns:**
>
> **To W1:** We would like to address your concern from the following two aspects:
>
> 1. From the embedding visualization (e.g., Figure 7), the embedding distributions of the pseudo-anomalous graphs are mostly separated into different regions from the normal data. As an illustrative example, Figure 7(j) shows that the pseudo-anomalous graphs successfully interpolate between the real anomalous and normal samples, which is good evidence to demonstrate the effect of the generated pseudo graph anomalies as the auxiliary signals to benefit decision boundary learning.
> 2. To more explicitly address your concerns, we evaluated SDM-NAT on AIDS (class 1) by employing the **normalized $k$-nearest-neighbor distance** to quantify the discrepancy. Specifically, we computed the average pair-wise one-nearest-neighbor distance between:
> * **Normal graphs vs. Normal graphs**, which is $0.0127$.
> * **Real anomalous graphs vs. Normal graphs**, which is $0.1056$.
> * **Pseudo-anomalous graphs vs. Normal graphs**, which is $0.0542$.
>
> The results confirm that the pseudo-anomalous graphs are sufficiently different from normal graphs, supporting their role in enhancing decision boundary learning.
>
> **To W2:** We would like to clarify how our approach ensures the generation of diverse pseudo-anomalous graphs:
>
> 1. Our method does not generate pseudo-anomalous graphs as a one-time, static process. Instead, the generation is dynamic and evolves continuously throughout training. Early in the process, the pseudo-anomalous graphs exhibit significant deviations from normal graphs, posing a relatively straightforward GLAD task. As training progresses, the generator will produce pseudo-anomalous graphs that increasingly resemble normal graphs, thereby escalating the difficulty of the task. This progressive shift aligns with the principles of curriculum learning [1], where the model begins with simpler examples and gradually tackles more challenging ones. As a result, the model can be exposed to a broad and diverse spectrum of pseudo-anomalous samples, rather than just a narrow subset.
> 2. In SDM-ATI and SDM-ATII, the adversarial interplay between generator and discriminator also drives diversity. The generator needs to vary its perturbation strategies to challenge the discriminator, preventing repetitive patterns. This results in the diversity of generated pseudo-anomalous graphs. In SDM-NAT, pseudo-anomalous graphs are generated by sampling from the latent distribution of normal data. Therefore, the generated pseudo-anomalies are expected to enclose the normal samples in the latent space, which naturally enhances the diversity. This also aligns well with our motivation illustrated in Figure 1.
> ```
> [1] Yoshua Bengio, et al. Curriculum Learning, ICML, 2009.
> ```
> **To W3:** The overlap observed in Figure 6 (top row) during training is a natural outcome of the adversarial design of SDM-ATII, where the generator produces pseudo-anomalous graphs resembling normal graphs to challenge the discriminator. Actually, while the overlap is observed, the majority of normal graphs and anomalous graphs are still separable, and it helps the model to learn an effective decision boundary. The results observed in the bottom row of Figure 6 fully validate this claim, where SDM-ATII indicates strong generalization by successfully distinguishing real anomalous graphs in the test phase. We will supplement this discussion in the paper.
>
> **To W4:** We agree with the reviewer that the robustness evaluation under data contamination is important. In response, we conducted an experiment on MUTAG (class 0) by injecting anomalous data into the training set at varying contamination levels defined as certain percentages of the normal data. The experimental results shown below indicate that the three SDM variants maintain stable performance under different contamination levels, which still outperform SOTA baseline DOH2SC (94.72\% AUC w/o contamination) under 10\% contamination level. This demonstrates the robustness and real-world applicability of our approach.
>
> $\begin{matrix}
> \\hline
> \text{Contamination Level}& &0\\%&10\\% &20\\% &30\\% \\\\\hline
> \text{SDM-ATI} & \text{AUC} &\bf95.83 (0.00) &94.77 (0.79) &93.85 (0.04) &94.23 (1.23)  \\\\
> & \text{F1-Score} &\bf83.33 (0.00) &82.61 (0.46) &80.24 (0.00) & 81.27 (1.96)\\\\\hline
> \text{SDM-ATII} & \text{AUC} & \bf99.31 (1.42) & 97.58 (0.48)&97.25 (0.02) &97.04 (0.15) \\\\
> & \text{F1-Score} &\bf99.13 (1.74) & 94.12 (0.00)&88.24 (0.00) &89.22 (0.98) \\\\\hline
> \text{SDM-NAT} & \text{AUC} & \bf100.00 (0.00)&97.67 (0.73) & 96.44 (0.21) &95.71 (2.17) \\\\
> & \text{F1-Score} &\bf100.00 (0.00) &92.16 (1.96) &88.24 (0.00) & 87.70 (3.39) \\\\\hline
> \end{matrix}$
>
> **To Other Comments:** We will (1) double-check and harmonize the format of "$\textit{i.e.,}$" and "$\textit{e.g.},$", and (2) supplement the ratio of samples in each category for all datasets.

---

> > ### Comment · Reviewer_1ZeV · 2025-04-02
> >
> > The authors’ responses have addressed my concerns. They provided convincing evidence regarding the diversity and effectiveness of the pseudo-anomalous graphs, clarified the distributional overlap in Fig. 6, and included results under data contamination. These additions further improve the clarity and credibility of this paper. Accordingly, I decide to raise the overall score.

---

> > > ### Author Response · Authors · 2025-04-03
> > >
> > > Dear Reviewer 1ZeV:
> > >
> > > Thank you for the positive feedback. We greatly appreciate your constructive comments to help us improve the paper and are glad that our responses can address your concerns.
> > >
> > > Best regards,
> > >
> > > Authors

---

### Official Review · Reviewer_5giQ · 2025-03-14

**Overall Recommendation:** 3

**Summary:**

The paper introduces a new framework called Self-Discriminative Modeling (SDM) for detecting anomalous graphs. The proposed method trains a deep neural network using only normal graphs, without access to real anomalous examples. To achieve this, the authors generate pseudo-anomalous graphs from normal graphs. These pseudo-anomalous graphs help the model learn an effective decision boundary for identifying real anomalies. The authors propose three versions of their method: two based on adversarial training (SDM-ATI and SDM-ATII) and one based on a simpler, non-adversarial approach (SDM-NAT). Experiments conducted on 12 benchmark datasets show that all three versions of SDM outperform existing methods, with the non-adversarial version (SDM-NAT) achieving the most stable and highest performance.

**Claims And Evidence:**

Overall, the claims in the paper are supported by the provided experiments. The authors show that the proposed Self-Discriminative Modeling (SDM) methods outperform existing techniques across multiple datasets.

**Essential References Not Discussed:**

Nothing as far as I know.

**Experimental Designs Or Analyses:**

I checked the experimental setup described in the Experiments section and the appendix. The authors include experiments in multiple settings, which I think is good. The baselines used by the authors are also sufficiently many. No major concern on the experimental design and analysis.

**Methods And Evaluation Criteria:**

The proposed methods and evaluation criteria make sense for the anomalous graph detection task. Generating pseudo-anomalous graphs to learn decision boundaries when real anomalies are unavailable is reasonable. The use of adversarial (SDM-ATI, SDM-ATII) and non-adversarial (SDM-NAT) approaches for generating pseudo-anomalous data also make sense. The evaluation metrics (AUC and F1-score) and chosen benchmark datasets are suitable, and commonly used in many previous works.

**Other Comments Or Suggestions:**

Some of the statements in the paper need clarification:
- ‘no overlap between D and \tilde{D}’. Could you clarify on this, as the distribution of graphs is continuous, there may be some overlap in the support of the distribution?
- ‘Note that this is an unsupervised learning problem, of which the training data do not contain any anomalous graphs’. In general, unsupervised learning problems only means that label is unavailable. However, in the labeled data, there could also be anomalous data that a model needs to discover. Please clarify the setup.

**Other Strengths And Weaknesses:**

Nothing additional.

**Questions For Authors:**

Please address the concerns above.

**Relation To Broader Scientific Literature:**

The paper builds clearly on existing methods like DeepSVDD-based approaches and GAN-based anomaly detection techniques. It addresses known issues of strong assumptions in graph embedding distributions in the previous works, by generating pseudo-anomalous graphs using various techniques. I think the contributions of the paper fills some of the gap in the previous literature.

**Theoretical Claims:**

No theorems provided in the paper, but the mathematical motivations behind the training objective make sense.

---

> ### Author Rebuttal · Authors · 2025-03-31
>
> **We thank the reviewer for the positive comments. Below are our responses to your concerns:**
>
> **To Q1:** The statement "no overlap between $\mathscr{D}$ and $\tilde{\mathscr{D}}$" is intended to facilitate a more precise problem definition for our paper, as we would not be able to identify normal or abnormal for a data point in the overlapping region between $\mathscr{D}$ and $\tilde{\mathscr{D}}$.  This is also a common assumption defined in anomaly detection literature [1], where normal and anomalous instances are modeled as originating from distinct distributions to establish a clear separation. However, we recognize the reviewer’s valid concern. In a continuous graph space, distributions may exhibit overlapping supports. We would like to answer this question by the following two aspects:
> 1. The "no overlap" assumption refers to the idealized setup where $\mathscr{D}$ and $\tilde{\mathscr{D}}$ are generated from non-overlapping processes. This abstraction facilitates the design of our anomaly detection method by providing a clean delineation between normal and anomalous behaviors.
> 2. In practice, real-world graph distributions may not be perfectly separable, which can be caused by outliers or the nature of the data. Our framework, however, is designed to handle such scenarios effectively. For instance, we generate pseudo-anomalous graphs to simulate $\tilde{\mathscr{D}}$, which may lie close to normal graphs in the feature space. This is evident in our t-SNE visualizations (e.g., Figures 7 and 8), where the proximity of these distributions tests the model’s ability to detect subtle anomalies. Our simulation experiment (in Sec 3.2) and empirical result (in Sec 3.3-3.5) on benchmark datasets demonstrate that the method performs robustly even when the theoretical assumption of non-overlapping supports is relaxed.
> ```
> [1] Lukas Ruff, et al. Deep One-Class Classification, ICML, 2018.
> ```
> **To Q2:** In our paper, we describe our approach as an unsupervised learning problem where "the training data do not contain any anomalous graphs." This means that during the training phase, the model is exposed exclusively to normal graphs from the distribution $\mathscr{D}$, with no anomalous graphs from $\tilde{\mathscr{D}}$ present. This setup aligns with the standard "one-class" anomaly detection paradigm [2], a well-established approach under unsupervised settings. Specifically:
> 1. Training Phase: The training dataset consists solely of normal graphs (from $\mathscr{D}$), and no labels are provided, consistent with unsupervised learning.
> 2. Testing Phase: The model is evaluated on a separate test set that includes both normal graphs (from $\mathscr{D}$) and anomalous graphs (from $\tilde{\mathscr{D}}$). The task is distinguishing anomalies from normal instances, despite not encountering anomalies during training.
>
> The reviewer is correct that, in a broader unsupervised learning context, training data could contain a mix of normal and anomalous instances without labels, requiring the model to discover anomalies implicitly. Therefore, we also conducted an experiment to evaluate our performance in the data contamination case (refer to the response to Reviewer 1ZeV's W4), where part of the anomalies are included in the training dataset. We can observe that our method also works when the training data have some unlabeled anomalies.
> ```
> [2] Bernhard Schölkopf, et al. Estimating the support of a high-dimensional distribution, Neural Computation, 2001.
> ```

---

### Decision · Program_Chairs · 2025-05-01

**Decision:**

Accept (poster)

**Comment:**

Three reviewers have evaluated this work and scored it 3, 3 and 4. Overall, all reviewers are satisfied with the rebuttal and advocate acceptance.